



# Atmospheric CO₂ dynamics in a coastal megacity: spatiotemporal
# patterns, sea-land breeze impacts, and anthropogenic-biogenic
# emission partitioning
Jinwen Zhang[1], Yongjian Liang[2], Chenglei Pei[2], Bo Huang[3], Yingyan Huang[2], Xiufeng Lian[1,3], Shaojie
Song[4], Chunlei Cheng[1], Cheng Wu[1], Zhen Zhou[1], Junjie Li[5], Mei Li[1*]
[1]College of Environment and Climate, Institute of Mass Spectrometry and Atmospheric Environment, Guangdong Provincial
Engineering Research Center for Online Source Apportionment System of Air Pollution, Guangdong-Hongkong-Macau Joint
Laboratory of Collaborative Innovation for Environmental Quality, Jinan University, Guangzhou 511443, China
[2]Guangzhou Ecological and Environmental Monitoring Center Station, Guangdong, Guangzhou 510006, China
[3]Guangzhou Hexin Instrument Co., Ltd., Guangdong, Guangzhou 510530, China
[4]College of Environmental Science and Engineering, Nankai University, Tianjin 300350, China
[5]School of Environment, Beijing Jiaotong University, Beijing 100044, China
*Correspondence to*: Mei Li (limei@jnu.edu.cn)
**Abstract.** Accurate quantification of urban carbon dioxide ($CO_2$) emissions is essential for evaluating the efficacy of urban
climate mitigation policies. However, the complex interplay of anthropogenic emissions, biogenic fluxes, and meteorological
processes in coastal megacities poses significant challenges to characterizing urban $CO_2$ dynamics. To address this, we present
an observation-based framework that integrates high-precision $CO_2$ monitoring, meteorological analyses, and $\Delta CO/\Delta CO_2$
ratios (Rco) to resolve spatiotemporal $CO_2$ variations, quantify sea-land breeze (SLB) effects, and partition anthropogenic and
biogenic contributions. Applied in Guangzhou, a coastal megacity, our approach captures a pronounced urban–rural gradient.
The coastal site shows the largest seasonal amplitude (25.63 ppm), resulting from wintertime transport of urban emissions and
summertime inflow of marine air. Diurnally, suburban $CO_2$ variations are dominated by biogenic activity (summer amplitude:
39.90 ppm), while urban signals reflect anthropogenic influence. SLB generally reduces coastal $CO_2$ by 5.87 ppm but leads to
a summer accumulation (+2.08 ppm) under stable, low-wind conditions with shallow boundary layers. Regression-derived
Rco values (urban: 7.45 ± 1.38 ppb ppm$^{-1}$) reflect improved combustion efficiency linked to clean-air policies. Importantly,
our combined observational, modeling, and Rco framework reveals that biogenic fluxes offset 60.17% of anthropogenic $CO_2$
emissions during summer afternoons. The framework is validated against emission inventories, Normalized Difference
Vegetation Index data, and independent studies, demonstrating its robustness. This study enhances process-oriented
understanding of coastal carbon cycling and underscores the integration of meteorological and biospheric dynamics in urban
$CO_2$ assessments.
## 1 Introduction
Atmospheric carbon dioxide ($CO_2$), the predominant anthropogenic driver of climate change, is accumulating at unprecedented



rates in human history (WMO, 2024). Future $CO_2$ increments will exert stronger warming effects than equivalent past increases
due to climate system feedbacks (He et al., 2023), making emission control imperative. Despite covering only 3 % of global
land, urban areas generate over 70 % of carbon emissions (Crippa et al., 2021), positioning them as critical climate mitigation
targets. As fundamental administrative units capable of implementing carbon-reduction strategies, cities require high-precision
$CO_2$ monitoring to validate policy efficacy against emission goals.

High-precision monitoring of atmospheric $CO_2$ emissions in coastal megacities is critically urgent due to their dual role as
economic powerhouses and climate vulnerability hotspots. These regions exhibit intensive industrialization, dense populations,
and high fossil fuel consumption, driving disproportionate emissions of greenhouse gases and air pollutants (Shan et al., 2020;
Li et al., 2022; Cai et al., 2023) that exacerbate climate change and health risks (Patz et al., 2014). All ten cities most impacted
by climate-amplified natural disasters are coastal, highlighting their acute vulnerability (Kumar, 2021). Characterizing urban
$CO_2$ dynamics remains challenging due to complex terrain, unique land-sea weather systems (particularly diurnal sea-land
breezes), heterogeneous vegetation distribution, and seasonal biogenic fluxes (Leroyer et al., 2014; Lei et al., 2024; Raciti et
al., 2014; Järvi et al., 2012). While single-site studies and aircraft measurements have quantified temporal $CO_2$ variations and
anthropogenic components (Wei et al., 2020; Pitt et al., 2022; Newman et al., 2013; Niu et al., 2016; Ishidoya et al., 2020),
they fail to resolve coastal spatiotemporal heterogeneity or disentangle meteorological, anthropogenic, and biogenic drivers
across diverse landscapes. Critical knowledge gaps persist despite multi-site analyses in cities like Boston, Portland, Helsinki,
and San Francisco linking spatial variability to meteorology and local sources (Briber et al., 2013; Rice and Bostrom, 2011;
Kurppa et al., 2015; Shusterman et al., 2018), and methods for marine background screening (Verhulst et al., 2017). Persistent
knowledge gaps critically constrain coastal carbon research: (1) systematic analyses of coastal-specific weather systems
(notably sea-land breezes) on $CO_2$ dynamics remain absent; (2) emission inventories persistently neglect quantitative
assessment of biogenic fluxes; and crucially, (3) biogenic contributions to urban carbon budgets continue to lack robust
constraints. This triad of limitations fundamentally impedes the mechanistic understanding of coastal megacity carbon cycles
and undermines evidence-based mitigation strategies.

Sea-land breeze (SLB) circulation, a ubiquitous mesoscale phenomenon in coastal zones driven by land-sea thermal contrast
(Chen et al., 2016; Shen et al., 2021), is intensifying under climate change: frequency has increased across 70 % of China's
coastline over five decades due to widening land-ocean temperature differentials (Huang et al., 2025). Although SLB
significantly influences coastal air quality (Nie et al., 2020; Zhao et al., 2022; Wang et al., 2023; Zheng et al., 2024), its impacts
on urban $CO_2$ dynamics are unexplored. Short-term modeling suggests that daytime coastal $CO_2$ measurements may be
contaminated by SLB-advected respired $CO_2$ from the vegetation during the previous night, potentially biasing emission



inversions (Ahmadov et al., 2007). Preliminary trajectory analyses hint at SLB's importance in diurnal greenhouse gas
variability (Verhulst et al., 2017), while SLB-mediated cooling/humidification may enhance carbon uptake in subtropical
mangroves (Zhu et al., 2021). Thus, investigating SLB-$CO_2$ interactions is imperative for advancing coastal carbon cycle
science.

Urban vegetation constitutes a significant carbon reservoir, yet its contribution to carbon sequestration is frequently
underestimated in urban carbon budgets (Davies et al., 2011; Raciti et al., 2014; Gough and Elliott, 2012).Urbanization-induced
microclimates—notably extended the urban growing seasons (Melaas et al., 2016a; Melaas et al., 2016b)—stimulate
aboveground biomass accumulation, yielding growth rates equivalent to some forests, while also enhancing vegetation net
ecosystem exchange (Zhao et al., 2012; Briber et al., 2013; Briber et al., 2015). Yet, observational quantification of urban-
scale biogenic fluxes remains scarce. While radiocarbon ($^{14}C$) measurements provide robust fossil/biogenic partitioning
(Turnbull et al., 2015; Niu et al., 2016; Berhanu et al., 2017; Wang et al., 2022), their cost and discontinuous sampling limit
applicability. Eddy covariance measurements quantify net $CO_2$ fluxes within tower footprints (typically 1–2 km radius), yet
the derived flux partitioning reflects only local-scale dynamics and cannot represent city-wide carbon exchange processes
(Velasco et al., 2013; Menzer and Mcfadden, 2017; Sugawara et al., 2021; Wu et al., 2022b). Inversion models incorporating
co-emitted tracers (e.g., CO and $NO_2$) and isotopes face challenges from prior emission uncertainties and computational
demands (Feng et al., 2024; Newman et al., 2016; Gómez-Ortiz et al., 2025).

To elucidate atmospheric $CO_2$ dynamics in coastal megacities and bridge critical knowledge gaps in urban carbon cycling, this
study investigates Guangzhou—a representative Chinese coastal megacity—as a living laboratory. Guangzhou—a subtropical
coastal megacity ranking fourth among Chinese cities in annual GDP (population: 18.83 million)—features 43.77 % green
coverage, a pronounced maritime climate, and frequent SLB events (Qiu and Fan, 2013a; Mai et al., 2024b). Its high-precision
greenhouse gas monitoring network (covering urban, suburban, and coastal zones) with > 1.5 years of observations provides
an ideal platform for this study.

In this study, we analyze spatiotemporal heterogeneity in urban $CO_2$ concentrations and dissect the differential roles of
anthropogenic emissions, biogenic fluxes, and meteorological processes across urban, suburban, and coastal landscapes. This
approach advances understanding of urban-scale drivers governing $CO_2$ distribution patterns. Further, we employ an
observation-based diagnostic method using meteorological tower data to quantify SLB impacts on $CO_2$ dynamics and their
underlying mechanisms, addressing a fundamental gap in coastal carbon cycle science. Complementing this, we develop a
parsimonious framework integrating observational data, atmospheric transport modeling, and $CO/CO_2$ emission ratios to



partition anthropogenic ($CO_2ff$) and biogenic ($CO_2bio$) fluxes. This cost-effective strategy enables robust assessment of
biogenic contributions to urban carbon budgets, validated against bottom-up emission inventories, the NDVI and independent
studies. The resulting methodological framework provides a transferable blueprint for global coastal megacity carbon research
and offers scientific foundations for evidence-based urban climate mitigation policy design and efficacy evaluation.
**2 Data and methodology**
**2.1 Observational sites**
The high-precision greenhouse gas monitoring network in Guangzhou is illustrated in Fig. 1. Three stations—Nansha (NS),
Panyu (PY), and Conghua (CH) are symmetrically distributed along the city's predominant south-north wind axis, representing
coastal, urban, and suburban atmospheric conditions, respectively. Site selection criteria are detailed in the Supplement. All
stations employ tower-based sampling at similar heights: NS and PY at 48 m, and CH at 40 m. Monitoring spanned from
January 1, 2023, to September 30, 2024. From PY, the straight-line distances to NS and CH are 54 km and 89 km, respectively.

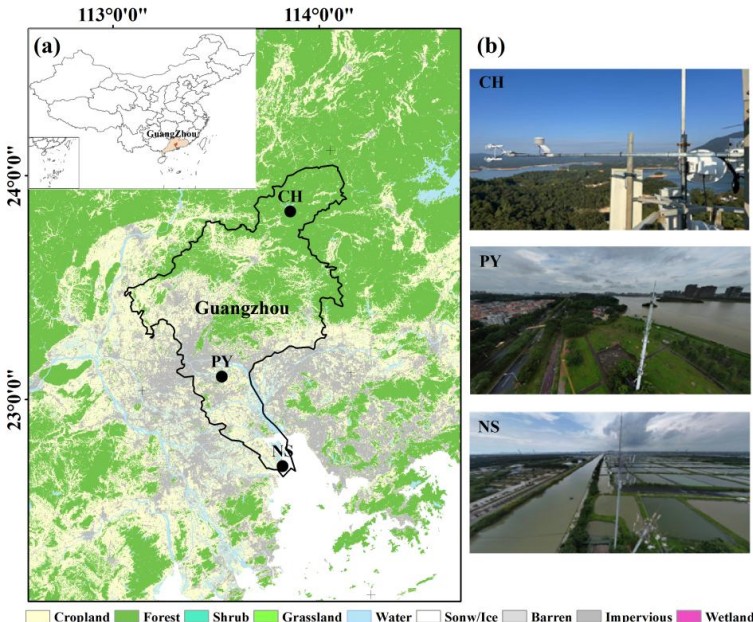


**Figure 1.** (a) Geographic locations of the NS, PY, and CH stations, with regional land use classification based on the 30 m resolution 2023
CLCD data (Yang and Huang, 2025). (b) Photograph of each station.
The NS station (113.63° E, 22.61° N) is located < 5 km from the coastline. This coastal site is surrounded by aquaculture ponds
and sparse wetlands. Infrastructure nearby includes the under-construction southern extension of Guangzhou Metro Line 18
(NW direction) and the S78 highway (2 km west). The PY station (113.38° E, 23.03° N) is situated in the densely populated
urban core, the tower is adjacent to Guangzhou University Town (north) and the Pearl River (south). A city road (100 m north)





and the S73 expressway (700 m west) contribute to local emissions. The CH station (113.78° E, 23.74° N) is positioned in the
northern suburbs. The site is bordered by subtropical evergreen broadleaf forests (north), a tourist resort (south), and a tea
processing plant. The G45 highway lies 3 km northwest. According to the 2023 EDGAR global emission inventory (Crippa et
al., 2024), grid-level $CO_2$ emissions for NS, PY, and CH are 3456, 15244, and 203 ton $km^{-2}$ $yr^{-1}$, respectively (Fig. S1 in the
Supplement).
**2.2 Monitoring system**
All three stations are equipped with similar monitoring systems, consisting of sampling modules, calibration modules, gas
analyzers, and data acquisition systems. Notably, the NS and PY stations utilize Picarro G2401 greenhouse gas analyzers to
measure $CO_2/CH_4/CO/H_2O$, with a $CO_2$ measurement precision of < 20 ppb (5 min, 1 σ). The CH station employs an ABB
GLA331-GGA greenhouse gas analyzer to measure $CO_2/CH_4/H_2O$, with a $CO_2$ measurement precision of < 25 ppb (5 min, 1
σ). $N_2O$ and CO are measured using a GLA351-N2OCM analyzer. Detailed monitoring system and principles of the
instruments are provided in the Supplement. Prior to field deployment, comparative tests were conducted in the laboratory to
ensure the analytical performance consistency of the instruments. Additionally, meteorological sensors (measuring wind speed,
direction, humidity, temperature, and pressure) are installed at the same height as the sampling inlets at NS and PY stations,
while the CH station lacks such sensors. Detailed descriptions of wind field characteristics at NS and PY stations are included
in the Supplement and illustrated in Fig. S2.
**2.3 Calibration methods**
The calibration module comprises two components: working standard curve establishment and target gas verification. High-
and low-concentration standard gases are used to establish calibration curves, while a mid-concentration standard gas is used
for target verification. The target and calibration gases are stored in inert-coated aluminum cylinders, uniformly supplied by
the China National Environmental Monitoring Center. All stations follow the same calibration protocol: (1) weekly calibration
curve establishment: high- and low-concentration gases are injected for 30 minutes each, with the final 5 minutes of instrument
response used for calibration; (2) target gas verification every 12 hours: mid-concentration gas is injected for 30 minutes, with
the final 5 minutes of response used for verification; (3) re-calibration is triggered if the residual value (H) from target
verification exceeds ± 0.2 ppm.

Calibration curves are derived from the instrument's response to calibration gas, yielding a linear calibration equation:
$Y = A \times X - B$ , (1)
where A, B are calibration coefficients. Calibrated $CO_2$ ($CO_{2,k}$) is calculated by:

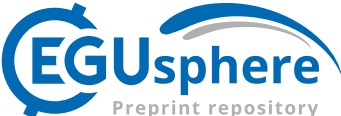

$CO_{2,k} = A \times CO_{2,m} - B$ ,                                         (2)
where $CO_{2,m}$ is the measured response. Daily 12-hour target gas verification is conducted to assess analyzer accuracy and
stability by calculating the residual H:
$H = (A \times CO_{2,c} - B) - CO_{2,n}$  .                                     (3)
where $CO_{2,n}$ is the standard $CO_2$ concentration of the target gas, where $CO_{2,c}$ is the analyzer response to the target gas.

To ensure high-precision and stable monitoring results, periods with $|H| \leqslant$  0.1 ppm are prioritized. Measurement
uncertainties for the analyzers at NS, PY, and CH stations, calculated as the standard deviation (SD) of H (Yang et al., 2021),
are 0.04, 0.02, and 0.04 ppm, respectively. In addition to daily calibration, maintenance personnel conduct weekly inspections
of instruments and station facilities, including checks on power supply stability, data logger functionality, and industrial control
computer status. Consumables (e.g., filters) are replaced as needed, and emergency repairs or instrument overhauls are
performed when necessary. Any instrument downtime caused by internal or external factors is documented in maintenance
logs, and affected data is flagged. Throughout the monitoring period, all three stations maintained data validity rates exceeding

154  90 %.

**2.4 Sea-land breeze identification**
The straight-line distances from the NS, PY, and CH stations to the coastline are 4, 58, and 130 km, respectively. The NS
station, closest to the coast, was selected to study Guangzhou's sea-land breeze (SLB) circulation. Prior to SLB identification,
local and background winds must be differentiated, as tower-measured winds (Fig. S2 in the Supplement) represent
superimposed local and background wind fields, where strong background winds can obscure SLB signals (Qiu and Fan,
2013b). The following equations distinguish background winds from local winds (Sun et al., 2022):
$U_b = \overline{\sum_{i=0}^{23} U_i}$ ,                                          (4)
$V_b = \overline{\sum_{i=0}^{23} V_i}$ ,                                          (5)
$U_l = U_O - U_b$ ,                                                              (6)
$V_l = V_O - V_b$ .                                                             (7)
where $U_O$ and $V_O$ are the observed wind fields from the tower, $U_b$ and $V_b$ denote background winds, and $U_l$ and $V_l$ represent
local winds.

A sea-land breeze day (SLBD) is defined as any 24 hours period exhibiting a distinct transition from sea breezes during the
day to land breezes at night (Xiao et al., 2023). SLB identification criteria vary regionally due to differences in topography



and coastline geometry (Huang et al., 2025). For the NS station, located north of the Pearl River Estuary, sea breeze and land
breeze directions are defined as 112–202° and 302–45°, respectively, based on local coastline features. Drawing from SLB
criteria in Table S1 in the Supplement and historical SLB patterns in the Pearl River Estuary (Qiu and Fan, 2013b; Zhang et
al., 2024; Mai et al., 2024b), the land breeze period is defined as 01:00–09:00, and the sea breeze period as 12:00–20:00.
Guangzhou's SLBD identification rules are: (1) daily mean wind speed < 10 m s$^{-1}$; (2) local winds persistently meet land/sea
breeze direction criteria for ≥ 4 hours within their respective periods, or (3) land/sea breeze directions occur for ≥ 4 hours
within any 5-hour window of their respective periods. Otherwise, the day is designated a non-SLB day (NSLBD).
**2.5 Estimation of CO$_2$tot, CO$_2$ff, and CO$_2$bio**
The observed CO$_2$ concentration enhancements at tall-tower sites represent the integrated effect of upwind surface fluxes
transported by atmospheric advection (Lin et al., 2003). Consequently, upwind carbon emissions can be inversely derived from
site-specific enhancement measurements coupled with their corresponding atmospheric footprints. We employ a simplified
methodology integrating observational data and atmospheric transport modeling to estimate total urban CO$_2$(CO$_2$tot) and CO
(COtot) emissions. This approach operates independently of a priori emission inventories for the study region, requiring only:
concentration enhancement observations at receptor sites, and simulated atmospheric footprints from transport models—
consistent with prior regional flux quantifications of CO$_2$, CH$_4$, and CO (Mitchell et al., 2018; Lin et al., 2021; Wu et al.,
2022a). CO$_2$tot and COtot are calculated as:
$$CO_2 tot = \frac{CO_{2,s}obs - CO_2 bg}{\sum_i Footprint_{i,s}} ,$$  (8)
$$COtot = \frac{CO_s obs - CObg}{\sum_i Footprint_{i,s}} ,$$  (9)
The numerators are hourly CO$_2$ and CO concentration enhancements ($\Delta CO_2$ and $\Delta CO$) at station s, where CO$_{2,s}$obs and CO$_s$obs
represent observed CO$_2$ and CO concentrations, while CO$_2$bg and CObg denote urban background concentrations (detailed in
the Supplement). The denominators are hourly total atmospheric footprints ($\sum_i Footprint_{i,s}$), where i denotes backward
particle release time from the receptor. Due to challenges in modeling mixed-layer depths during nighttime, morning, and
evening, flux analysis focuses on afternoon hours (12:00–16:00) (Boon et al., 2016; Mitchell et al., 2018; Lin et al., 2021).
Daily-scale CO$_2$tot and COtot are derived by dividing the mean afternoon $\Delta CO_2$ and $\Delta CO$ by the corresponding mean
$\sum_i Footprint_{i,s}$, The footprint quantifies the sensitivity of concentration enhancements at the observation site to upwind surface
fluxes, as detailed in Sect. 2.5.1. $\triangle CO_2$ is in units of [ppm], while footprint is in [ppm / (μmole m$^{-2}$ s$^{-1}$)], so CO$_2$tot, the
quotient between the two quantities, is in flux units of [μmole m$^{-2}$ s$^{-1}$].

Anthropogenic CO$_2$ emissions (CO$_2$ff) are derived from COtot and the CO/CO$_2$ emission ratio (R$_{CO}$), where R$_{CO}$ is determined



from real-time tower-measured data, as described in detail in Sect. 3.4:
$CO_2ff = \dfrac{COtot}{Rco}$, (10)
Biogenic fluxes ($CO_2bio$) are calculated as residuals:
$CO_2bio = CO_2tot - CO_2ff$. (11)
Positive $CO_2bio$ values indicate biogenic carbon emissions, while negative values denote carbon uptake, reflecting the dual
role of urban biospheres as $CO_2$ sources and sinks (Kim et al., 2025).

**2.5.1 Atmospheric transport model**

To trace airmass sources entering the urban domain and reaching observation sites, and to assess $CO_2$ emissions corresponding
to observed concentration enhancements, the Stochastic Time-Inverted Lagrangian Transport model (STILT-Rv2) was
employed for atmospheric transport simulations, driven by meteorological fields from the Weather Research and Forecasting
Model (WRFv4.1.1). In this study, STILT serves two purposes: (1) providing airmass trajectories for identifying marine
background concentrations in Guangzhou, and (2) generating atmospheric footprints for quantifying total $CO_2$ and CO
emissions.

The STILT model simulates atmospheric transport by releasing a set of air particles backward in time from the receptor location
at the observation height. These particles are tracked spatially and temporally as they disperse upwind. The resulting
trajectories delineate source regions influencing the receptor site and quantify the sensitivity of observed concentrations to
upwind surface fluxes, termed "source-receptor relationships" or "atmospheric footprints" (Lin et al., 2003; Fasoli et al., 2018).
Footprints represent the contribution of upwind sources/sinks to downwind concentration changes, with higher sensitivities
near receptors or under stable wind conditions, where boundary layer airmasses interact more directly with surface fluxes (Wu
et al., 2022a). For this study, 500 particles were released at 48 m (PY stations) heights, and traced backwards in time for 72 h.
Footprints were computed at 0.08° × 0.08° spatial resolution. Periods with total footprint sensitivities ($\sum_i \text{Footprint}_i$) below
the 10th percentile were excluded, indicating low sensitivity to regional surface fluxes (Lin et al., 2021).

**2.5.2 Uncertainty sources**

Uncertainties associated with emission estimates derived from Eqs. (12)–(14) primarily arise from: (1) observational
uncertainties, (2) background concentration uncertainties, (3) atmospheric transport uncertainties (footprints) and (4) $R_{CO}$
uncertainties. Here, footprint uncertainties are neglected under the assumption of unbiased atmospheric transport during
observations. The $R_{CO}$ uncertainty originates from factors (1) and (2). Collectively, the dominant uncertainties in $CO_2tot$ and



$CO_2$bio quantification stem from observational and background concentration errors, calculated as:
$E_u^2 = OBS_{u,c}^2 + BG_u^2$ ,                                                                                                    (12)
where $OBS_{u,c}$ represents uncertainty in urban atmospheric observations, and $BG_u$ represents uncertainty in urban background
concentrations. We cannot accurately quantify all error sources involved in instrumental measurements; some minor error
sources (e.g., uncertainty related to water vapor) may be negligible, while the primary uncertainty originates from discrepancies
between measured the concentration of air samples and calibration standards (Verhulst et al., 2017). Here, $OBS_{u,c}$ is calculated
as the standard deviation (SD) of residuals H (Yang et al., 2021). For urban background uncertainties:
$BG_{u,co2}^2 = CT_{co2,r}^2 + CT_{co2,s}^2$ ,                                                                                       (13)
$BG_{u,co}^2 = OBS_{co,r}^2 + OBS_{co,s}^2$ .                                                                                         (14)
$CO_2$ background uncertainty ($BG_{u,co2}$) combines the absolute monthly smoothed residuals ($CT_{co2,r}$) and variability (SD) of
monthly $CO_2$ concentrations ($CT_{co2,s}$) from Carbon Tracker (CT). Similarly, CO background uncertainty ($BG_{u,co}$) is derived
from monthly smoothed residuals ($OBS_{co,r}$) and variability ($OBS_{co,s}$) of in situ observations.
**3 Results and discussion**
**3.1 Spatiotemporal patterns of atmospheric $CO_2$**
Figure 2 presents the hourly mean time series of atmospheric $CO_2$ concentrations at the NS, PY, and CH stations in Guangzhou
from January 1, 2023, to September 30, 2024. To assess wind field impacts on $CO_2$ variability (Fig. S2), concentrations were
classified into five categories: local (wind speed < 1.5 m s$^{-1}$) and four directional sectors (wind speed ≥ 1.5 m s$^{-1}$) defined
as NE (0–90°), SE (90–180°), SW (180–270°), and NW (270–360°). All stations exhibited significant temporal variability,
with standard deviations (SD) of 13.90 (NS), 15.92 (PY), and 16.05 ppm (CH), consistent with urban and suburban
observations in Hangzhou, Beijing, Xi'an, and Seoul (Park et al., 2021; Yang et al., 2021; Chen et al., 2024; Liu et al., 2025).
PY showed the highest $CO_2$ levels and variability, driven predominantly by local-type emissions under low wind speeds. In
contrast, the NS coastal station exhibited elevated concentrations during northerly winds (NW/NE). Seasonal wind effects
were pronounced: summer southerly winds (SW/SE) reduced $CO_2$ at PY and NS (most notably at NS near the coast), while
winter northerly winds (NW/NE) increased $CO_2$ at NS.



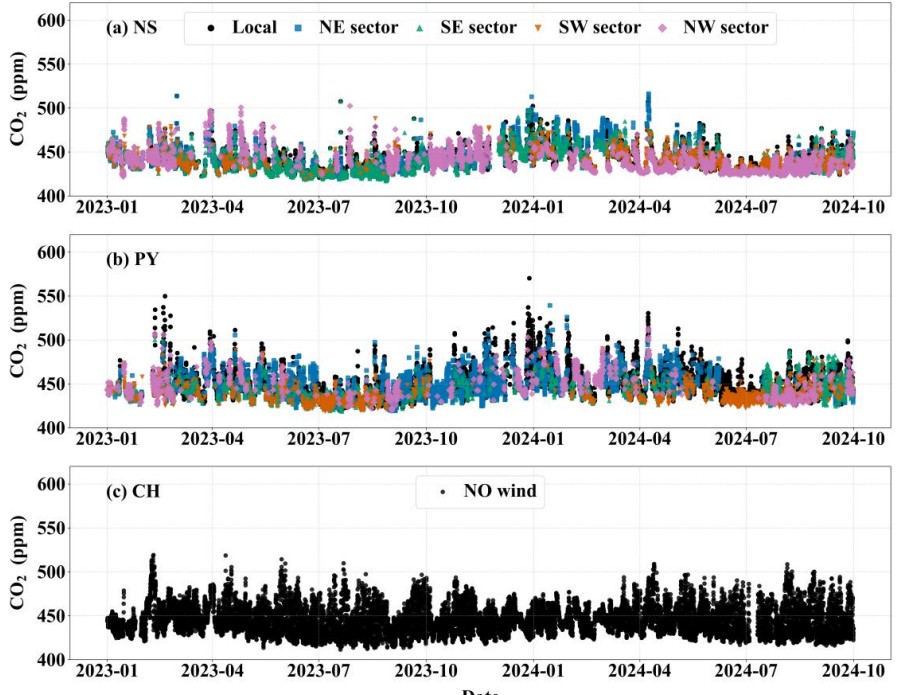

**Figure 2.** Time series of atmospheric $CO_2$ concentrations at the (a) NS, (b) PY, and (c) CH stations. For the NS and PY stations, data are color-coded by wind speed and direction, classified as either local (wind speed < 1.5 m s$^{-1}$) or one of four directional sectors (NE (0–90°), SE (90–180°), SW (180–270°), NW (270–360°)) for winds with speed ≥ 1.5 m s$^{-1}$.

Urban-rural $CO_2$ gradients vary globally due to differences in economic activity, population density, land use, and energy infrastructure, reflecting heterogeneous urban carbon emissions (Gao et al., 2022). In Guangzhou, mean $CO_2$ concentration differences between PY and NS/CH were 6.67 and 3.43 ppm, respectively, forming a distinct "urban dome" (urban > suburban > coastal). The NS-CH difference (3.44 ppm) highlights comparable gradients between suburban and coastal zones. This gradient mirrors Los Angeles's coastal megacity profile but with a smaller magnitude (Verhulst et al., 2017). Guangzhou's urban-suburban difference (3.43 ppm) aligns with Hangzhou's 2021 observations (4.96 ppm) (Chen et al., 2024) but is lower than Nanjing (8.1 ppm, 2014) and Beijing (12.4 ppm, 2018–2019) (Gao et al., 2018; Yang et al., 2021). It remains far smaller than Shanghai (55.1 ppm, 2014) and Baltimore (66.0 ppm, 2002–2006) (Pan et al., 2016; George et al., 2007). Over time, urban emissions may stabilize as suburban populations and fossil fuel demand grow, potentially narrowing urban-suburban $CO_2$ differences (Mitchell et al., 2018). For instance, Hangzhou's reduced gradient reflects urbanization-driven energy consumption, where suburban monitoring captures urban emission influences (Chen et al., 2024).

### 3.1.1 Seasonal variability of atmospheric $CO_2$

Figure 3 illustrates the monthly mean variations in atmospheric $CO_2$ concentrations at the NS, PY, and CH stations in



Guangzhou, alongside their correlations with the Normalized Difference Vegetation Index (NDVI). NDVI data at 1 km × 1 km
spatial resolution were obtained from NASA's EOSDIS Land Processes Distributed Active Archive Center (Didan, 2015), with
values within a 3-km radius buffer around each station center used for comparative analysis. All three stations exhibited
consistent seasonal $CO_2$ patterns, with higher concentrations in winter/spring and lower values in summer/autumn, mirroring
observations in Hangzhou (Chen et al., 2024). These variations arise from the combined effects of (1) seasonal biogenic flux
cycles, (2) anthropogenic emission variability, and (3) boundary layer height dynamics (Xueref-Remy et al., 2018). Enhanced
vegetation photosynthesis during warmer months (summer/autumn, Table S2 in the Supplement) strengthens biogenic carbon
sinks, while higher boundary layer depths (Fig. S6 in the supplement) and southerly marine air masses (Fig. 2) promote
atmospheric mixing and $CO_2$ dispersion.

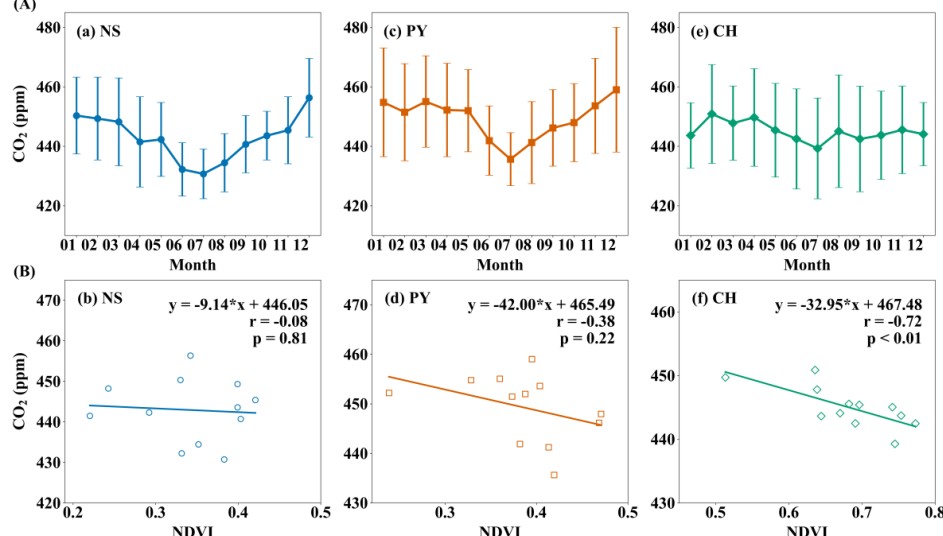


**Figure 3.** (A) Variations in monthly mean $CO_2$ concentrations and (B) their correlations with Normalized Difference Vegetation Index

(NDVI) for the (a–b) NS, (c–d) PY, and (e–f) CH stations. Error bars indicate ± 1 standard deviation (SD).

The amplitudes of the seasonal variation of $CO_2$ at NS, PY, and NS are 25.63, 23.38, and 11.59 ppm, respectively. NS and PY
peaked in December and troughed in July, whereas CH peaked in February and troughed in July. NS's large amplitude reflects
its extreme December highs and July lows. In December, prevailing northerly winds (Figs. 2a and S2a) transported urban
emissions to downwind NS, narrowing its $CO_2$ difference with PY to 2.68 ppm. Conversely, July saw NS's $CO_2$ concentrations
fall to the lowest among all stations—4.93 ppm and 8.56 ppm below PY and CH, respectively—establishing a south-to-north
increasing gradient (coastal < urban < suburban). This gradient aligns with marine-influenced southerly air masses, which
dilute coastal $CO_2$ while transporting urban emissions northward, potentially accumulating $CO_2$ in northern suburbs.

Despite CH's stronger biogenic coupling (NDVI correlation: −0.72; Fig. 3f), NS's $CO_2$ levels remained 9.80 ppm lower than





CH in summer and 5.80 ppm higher in winter, underscoring transport-dominated over biogenic controls at the coastal site.
NS's weak NDVI correlation (−0.08) and low NDVI range (0.22–0.42) further support minimal biogenic influence.
Additionally, deeper summer boundary layers at NS and PY (Fig. S6) enhanced vertical $CO_2$ dispersion. In February, CH
recorded its annual $CO_2$ maximum, driven by vegetation respiration during early growth stages and elevated emissions from
fireworks around the Lunar New Year, as CH's location falls outside Guangzhou's fireworks restriction zones
(https://www.gz.gov.cn/gfxwj/qjgfxwj/chq/qf/content/post_7198980.html, last access: 18 June 2025). This finding is
corroborated by CO observations: CH's CO concentrations peaked in February due to firework emissions, whereas other sites
peaked in December (Fig. S7 in the Supplement).
**3.1.2 Diurnal variations of atmospheric $CO_2$**
The diurnal patterns of atmospheric $CO_2$ concentrations at NS, PY, and CH stations in Guangzhou consistently exhibited lower
daytime and higher nighttime values (Fig. 4). This is attributed to the shallow nocturnal boundary layer, which traps
anthropogenic and biogenic emissions near the surface, elevating $CO_2$ levels. After sunrise, surface heating deepens the
boundary layer, diluting surface emissions and entraining free tropospheric air with lower $CO_2$ concentrations. Concurrently,
daytime photosynthetic uptake further reduces near-surface $CO_2$ (Mitchell et al., 2018). We further evaluate urban-suburban-
coastal differences in these processes. At PY, the $CO_2$ peak occurred at 08:00–09:00, aligning with morning traffic peaks,
reflecting dominant anthropogenic influences. CH's peak appeared 1–2 hours earlier than PY due to its longitudinal and
elevational position, where earlier sunrise accelerates the breakup of the nocturnal stable boundary layer. Both PY and CH
reached minima at 16:00–17:00, likely linked to afternoon photosynthetic activity. NS exhibited irregular peak/valley timing.

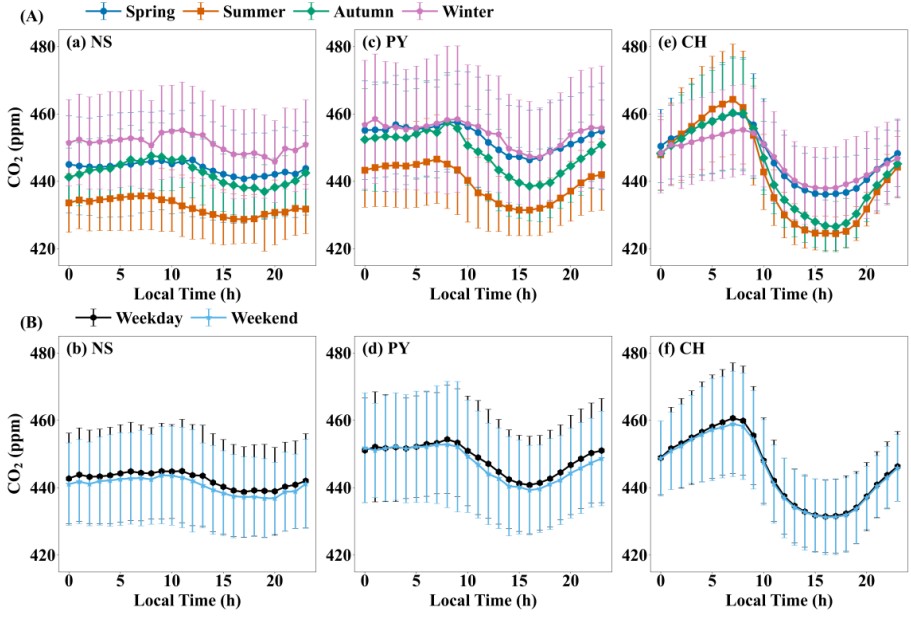




**Figure 4.** Diurnal $CO_2$ variations at the (a–b) NS, (c–d) PY, and (e–f) CH stations across (A) seasons and (B) weekdays/weekends.
Seasons are defined as spring (Mar–May), summer (Jun–Aug), autumn (Sep–Nov), and winter (Dec–Feb). Error bars indicate ± 1 SD.
Diurnal amplitudes at CH and PY were larger in summer/autumn than winter/spring, driven by vegetation activity and
boundary layer dynamics (Fig. S8 in the Supplement). Summer/autumn conditions in Guangzhou—abundant light, warmth,
and rainfall—optimize vegetation growth, enhancing daytime photosynthesis and nighttime respiration (Dusenge et al., 2019).
Optimal canopy temperatures for subtropical evergreen forests (~30 °C) (Liu et al., 2015) align with CH/PY's summer/autumn
daytime temperatures (Table S2), explaining their amplified amplitudes. However, the diurnal amplitude of $CO_2$ at CH in
summer and autumn is 2.63 times and 1.77 times that at PY, respectively. The diurnal amplitude of atmospheric $CO_2$
concentration at CH in summer is 39.90 ppm, which is close to the diurnal amplitude of $CO_2$ concentration in the suburbs of
Hangzhou in summer (35.29 ppm)(Chen et al., 2024). Despite similar temperatures, CH's larger NDVI range and stronger
NDVI-$CO_2$ correlation (−0.72 vs. PY; Figs. 3d and f) highlight greater biogenic dominance, with pronounced daytime uptake
and nighttime respiration. NS showed the smallest diurnal amplitudes across seasons (e.g., 5.60 ppm in summer), attributable
to sparse vegetation (low NDVI: 0.22–0.42) and frequent summer southerly marine air masses, which dilute coastal $CO_2$.

Figure 4B contrasts weekday-weekend diurnal $CO_2$ patterns. All stations showed higher weekday concentrations, diverging
from Hangzhou and Beijing (Yang et al., 2021; Chen et al., 2024) but aligning with Paris and Boston (Briber et al., 2013;
Xueref-Remy et al., 2018). At CH, smaller daytime weekday-weekend differences suggest biogenic fluxes outweigh
anthropogenic variations. At PY, reduced weekend traffic (central urban location) drove daytime declines. NS's persistently
higher weekday $CO_2$ reflects construction activities (e.g., Metro Line 18 extension) and weekday-intensive port operations
(e.g., Nansha Container Terminal Phase III, 5 km east), which maintain 24/7 workflows (e.g., Shanghai Port's daily operational
indices: http://sisi-smu.org/2025/0422/c12041a243211/page.htm, last access: 18 June 2025).
**3.2 Sea-land breeze impacts**
Based on meteorological observations from the NS coastal tall tower, 84 sea-land breeze days (SLBD) were identified in
Guangzhou between January 2023 and September 2024, accounting for 13.14 % of the monitoring period, with peaks in spring
and autumn. These transitional seasons between summer and winter are characterized by weaker synoptic systems and lighter
background winds, favoring SLBD occurrence (Mai et al., 2024b). Our results align with SLBD seasonal distributions for the
Pearl River Estuary cities of Zhuhai and Guangzhou in 2022 (Zhang et al., 2024; Mai et al., 2024b). Figure 5 compares $CO_2$
concentrations during SLBD and non-SLB days (NSLBD) across stations. Overall, average $CO_2$ concentrations during SLBD
were 5.87 ppm (NS), 3.08 ppm (PY), and 0.75 ppm (CH) lower than during NSLBD, indicating that sea-land breeze (SLB)
circulation enhances $CO_2$ dispersion, with coastal > urban > suburban impacts, similar to SLB-driven $PM_{2.5}$, $PM_{10}$, and ozone





dispersion patterns in Tianjin (Hao et al., 2017). Seasonal differences were pronounced: SLB promoted $CO_2$ dispersion at NS
and PY in spring, winter, and autumn (spring > winter > autumn), but increased $CO_2$ accumulation in summer. Tianjin similarly
observed summer $PM_{2.5}/PM_{10}$ accumulation under SLB (Hao et al., 2017). At CH, SLBD reduced $CO_2$ in spring, summer, and
autumn but increased it in winter, likely due to limited inland SLB penetration and competing winter processes.

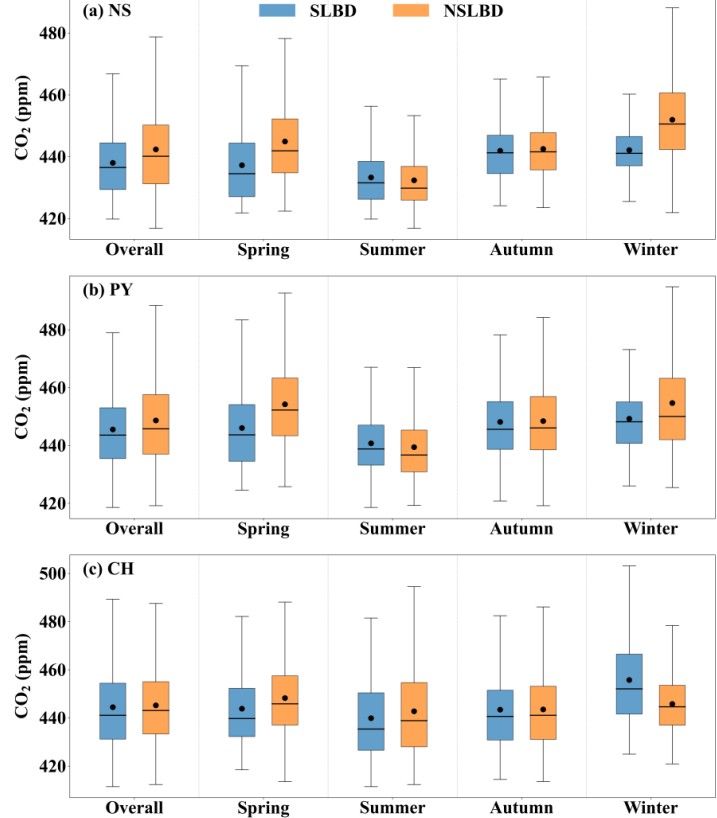


**Figure 5.** Boxplots of atmospheric $CO_2$ concentrations (black dots denote means) during sea-land breeze days (SLBD) and non-SLB
days (NSLBD) by station and season, with outliers excluded.
To resolve seasonal and diurnal SLB impacts, we analyzed $CO_2$ diurnal variations during SLBD and NSLBD (Fig. 6). Focusing
on NS (due to similar PY-NS trends and space constraints), spring and winter SLBD reduced $CO_2$ concentrations by 7.76 ppm
and 9.77 ppm (hourly mean differences), respectively, driven by stronger winds (Fig. 6) and deeper boundary layers (Fig. S9
in the Supplement). Autumn SLB only reduced $CO_2$ during sea breeze hours (mean difference: 1.69 ppm). Autumn's weaker
winds and boundary layers resulted in reduced dispersion compared to spring/winter. In summer, SLB increased $CO_2$ by 2.08
ppm (sea breeze hours) due to stable atmospheric stratification. Summer temperatures were 6.00 °C and 12.19 °C higher than
spring and winter (Table S2), respectively. Under calm, rain-free conditions, the collision of moist marine air with dry-hot
coastal land formed a thermal internal boundary layer (TIBL), inducing low-level temperature inversions near the SLB



convergence zone (Liu et al., 2001; Reddy et al., 2021). These inversions suppressed horizontal/vertical mixing, trapping $CO_2$
(Stauffer et al., 2015; Hao et al., 2024). NS's summer SLBD winds averaged 1.05 m s$^{-1}$ (sea breeze) and 0.96 m s$^{-1}$ (land
breeze)—38.60 %, 63.16 %, and 15.32 % lower than spring, winter, and autumn winds, respectively—while boundary layer
heights (590.54 m) were 9.51 % shallower than NSLBD (Fig. S9). Weak winds and shallow boundary layers stabilized
atmospheric stratification, limiting $CO_2$ dispersion and elevating ground-level $CO_2$ by up to 4.03 ppm.

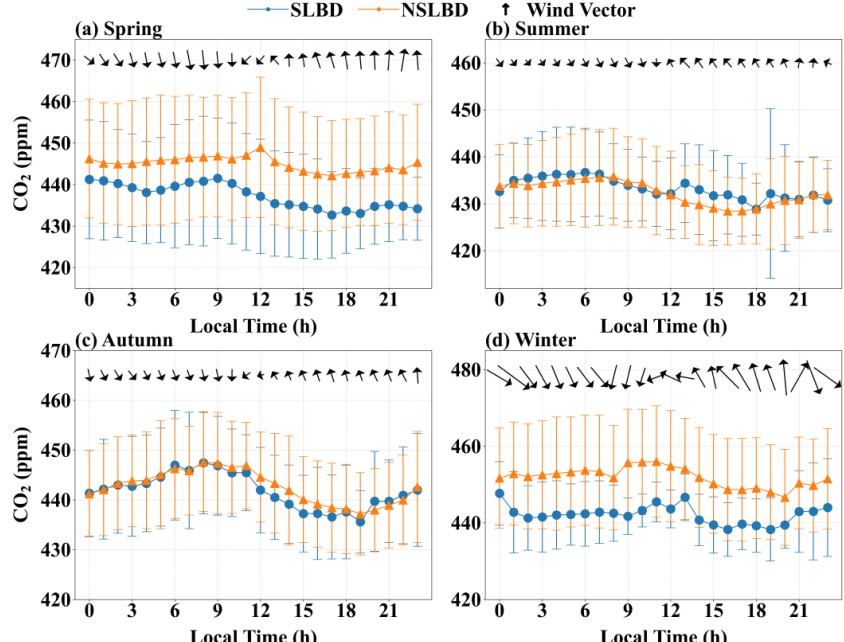


**Figure 6.** Diurnal variations in $CO_2$ concentrations, wind direction, and wind speed at the coastal station (NS) during sea-land breeze
days (SLBD) and non-SLB days (NSLBD) by season. Error bars indicate ± 1 SD.
**3.3 $CO_2$ enhancements and uncertainties**
Figure S10 (in the Supplement) presents the time series of observed $CO_2$ and CO concentrations at Guangzhou's stations
relative to marine backgrounds from January 1 to December 27, 2023. Compared to urban observations with significant hourly
variability, marine background concentrations in Guangzhou remained stable, with summer and winter $CO_2$ standard deviations
of 0.94 ppm and 0.67 ppm, respectively, indicating minimal local source/sink influences. Using Eqs. (13) and (14), marine
background uncertainties were calculated (Table S3 in the Supplement). Summer and winter $CO_2$ marine background
uncertainties were 0.96 ppm and 0.70 ppm, respectively, constraining urban marine background uncertainties below 1 ppm—
slightly lower than Los Angeles's 1.4 ppm (Verhulst et al., 2017). CO marine background uncertainties were 12.68 ppb
(summer) and 18.36 ppb (winter).

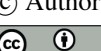



Based on marine backgrounds, CO₂ enhancements were derived for all stations. Figure 7 shows enhancements across all hours,
afternoon (12:00–16:00), and midnight (00:00–05:00) periods in 2023, summer, and winter. Annual median enhancements
were 13.59 (NS), 17.70 (PY), and 16.29 ppm (CH), with pronounced spatiotemporal variability—closely aligning with the 10–
20 ppm range observed annually in the Beijing-Tianjin-Hebei (BTH) urban cluster of China (Han et al., 2024). In summer,
enhancements followed a south-to-north gradient: 7.00 (NS), 13.23 (PY), and 16.91 ppm (CH). Afternoon enhancements
peaked at PY (6.92 ppm), critical for emission inversion, while midnight enhancements at CH reached 31.36 ppm—1.85 times
and 3.43 times higher than PY and NS. Winter afternoon enhancements reversed this pattern: 16.58 (NS), 12.37 (PY), and 7.45
ppm (CH), with NS and PY values 4.39 times and 1.79 times higher than summer. Midnight enhancements at CH remained
highest in winter (18.87 ppm), despite a 38.93 % reduction from summer.

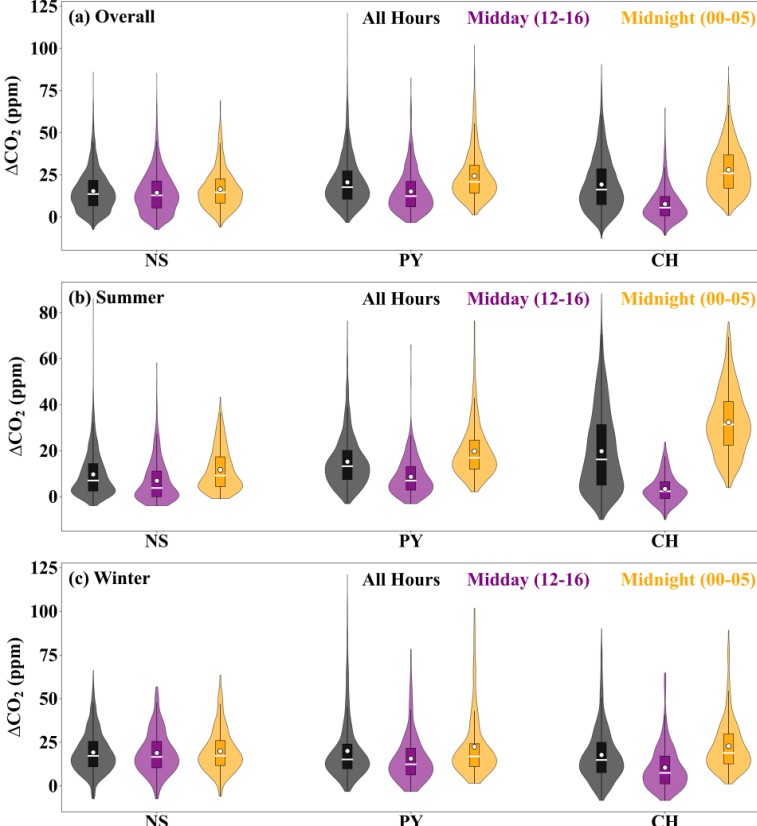


**Figure 7.** Hourly CO₂ enhancement above the marine background level at each station during the (a) overall, (b) summer, and (c)

winter periods. The white dots represent the mean values.

This spatiotemporal variability reflects divergent influences of anthropogenic emissions, biogenic fluxes, and atmospheric
mixing. At CH, strong diurnal shifts in enhancements (e.g., 31.36 ppm summer midnight) highlight biogenic dominance, with
long-tailed distributions (Fig. 7). Stable, shallow nighttime boundary layers trapped respiratory emissions near the surface,



consistent with isotopic studies in Xi'an (32.80 ppm) and Switzerland (30.00 ppm) (Wang et al., 2021; Berhanu et al., 2017).
At NS, transport dominated: summer southerly marine air masses reduced enhancements, while winter northerly winds
transported urban emissions downstream, raising NS enhancements to PY levels (exceeding PY in afternoons). PY's
enhancements were primarily anthropogenic, validated by CO co-variation. CO, a tracer for combustion-derived $CO_2$
(Newman et al., 2013; Che et al., 2022), showed significantly higher concentrations at PY (Fig. S10). PY's median midnight
CO enhancements in summer were 2.04 times and 1.43 times higher than NS and CH (Fig. S11 in the Supplement). Shallow
nocturnal boundary layers localized anthropogenic CO near the surface, with minimal vertical/horizontal transport, confirming
PY's anthropogenic dominance.
**3.4 Continuous observations of $\Delta CO/\Delta CO_2$ ratios**
Reduced Major Axis regression (Model II) was applied to analyze the relationship between CO ($\Delta CO$) and $CO_2$ ($\Delta CO_2$)
concentration enhancements across stations, with the $\Delta CO/\Delta CO_2$ ratio ($R_{CO}$) derived from regression slopes (Fig. 8). In 2023,
$R_{CO}$ values for NS, PY, and CH were $8.48 \pm 1.81$, $7.45 \pm 1.38$, and $4.16 \pm 3.59$ ppb ppm$^{-1}$, respectively, with correlation
coefficients of 0.78, 0.72, and 0.33, indicating significant spatiotemporal heterogeneity. Summer $R_{CO}$ was generally lower than
winter, with CH exhibiting the lowest seasonal value ($2.66 \pm 0.83$ ppb ppm$^{-1}$). Winter maxima occurred at NS ($11.03 \pm 1.49$
ppb ppm$^{-1}$), followed by PY ($9.08 \pm 1.13$ ppb ppm$^{-1}$) and CH ($8.56 \pm 1.82$ ppb ppm$^{-1}$).

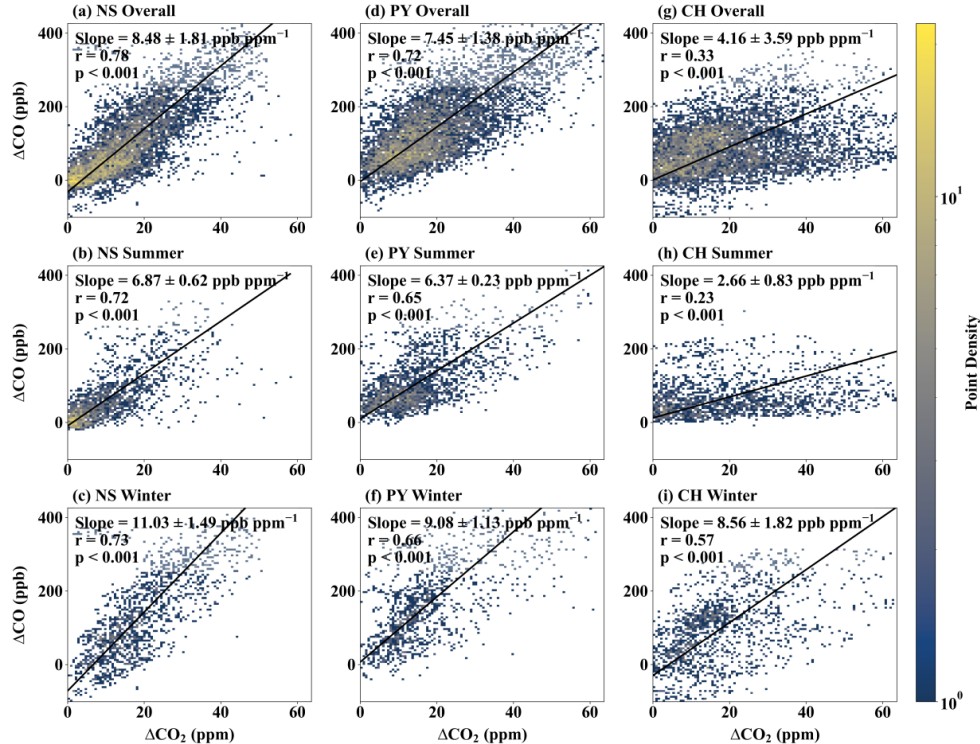


**Figure 8.** Relationships between $\Delta CO_2$ and $\Delta CO$ concentration enhancements analyzed by geometric mean regression for different

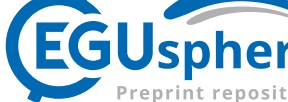



seasons at the (a–c) NS, (d–f) PY, and (g–i) CH stations, where the slope values represent the $\Delta CO/\Delta CO_2$ ratios ($R_{CO}$).
Comparatively, Beijing's urban $R_{CO}$ in 2019 was measured at $10.46 \pm 0.11$ ppb ppm$^{-1}$ using portable Fourier-transform
spectroscopy (Che et al., 2022), while Shanghai and Los Angeles showed $10.22 \pm 0.40$ and $9.64 \pm 0.46$ ppb ppm$^{-1}$, respectively,
based on satellite and model data (Wu et al., 2022a). Guangzhou's lower $R_{CO}$ reflects improved combustion efficiency driven
by stringent post-2013 air quality policies. For example, Beijing's $R_{CO}$ dropped from $> 30$ ppb ppm$^{-1}$ in 2006 (Han et al., 2009)
to $10.22 \pm 0.40$ ppb ppm$^{-1}$ by 2020 (Wu et al., 2022a), with similar declines during the 2008 Olympics and 2020 COVID-19
lockdowns (Wang et al., 2010; Cai et al., 2021). In Guangdong, policies restricting coal plants, retiring inefficient industries,
and promoting electric vehicles further enhanced combustion efficiency. For instance, attributed to strict air pollution controls,
$SO_2$ and $NO_2$ levels in this region decreased by 85 % and 35 % respectively in 2019 compared to 2006 (Hu et al., 2021),
demonstrating improved fossil fuel combustion efficiency. Mai et al. (2021) also reported the combustion efficiency
improvement in the Pearl River Delta region due to technological advancements in gasoline vehicles.

Seasonal $R_{CO}$ variations stem from biogenic flux and transport dynamics. Summer's weaker $\Delta CO$-$\Delta CO_2$ correlations at CH
(Fig. 3B) reflect dominant biogenic influences (daytime uptake and nighttime respiration), as reported in Beijing, Indianapolis,
and Switzerland (Turnbull et al., 2015; Berhanu et al., 2017; Che et al., 2022). Biogenic impacts decreased from suburban >
urban > coastal, aligning with vegetation gradients. Winter's higher $R_{CO}$ at CH and NS correlated with reduced biogenic
activity and northerly transport of urban emissions under stable boundary layers. Berhanu et al. (2017) attributed
winter $R_{CO}$ increases to cold-air advection and boundary layer accumulation. NS's winter $R_{CO}$ (4.16 ppb ppm$^{-1}$ higher than
summer) linked to urban airmass origins, while PY's seasonal shifts reflected suburban source-sink variations. Although
secondary CO from upwind Volatile Organic Compounds (VOCs) and $CH_4$ oxidation could perturb $R_{CO}$, their combined
contribution was merely 1 % in coastal urban regions (Griffin et al., 2007).
**3.5 Partitioning anthropogenic and biogenic fluxes**
Given the CH station's heightened sensitivity to biogenic fluxes—particularly during summer when its $R_{CO}$ incorporates more
pronounced biogenic components—contrasted with NS station's dominant atmospheric transport influences (especially winter
$R_{CO}$ perturbations from upwind urban emissions), PY station was selected to quantify total $CO_2$ emissions ($CO_2$tot) driving
observed concentration enhancements. Utilizing site-specific $R_{CO}$ measurements, $CO_2$tot was partitioned into fossil-derived
$CO_2$ ($CO_2$ff) and biogenic $CO_2$ ($CO_2$bio).

Fig. 9 displays mean $CO_2$tot, $CO_2$ff, and $CO_2$bio at PY station during summer and winter afternoons (12:00–16:00), with error
bars denoting seasonal standard deviations. Persistent dominance of $CO_2$ff over $CO_2$bio underscores anthropogenic control of



urban $CO_2$ emissions—a pattern corroborated in Chinese cities like Beijing (65 ± 3 %) and Xi'an (82 ± 2 %) where fossil fuels
dominate observed enhancements (Wang et al., 2022). Elevated winter (December) $CO_2ff$ relative to summer (July) primarily
reflects atmospheric dynamics dominating summertime dilution: marine air mass advection and boundary layer changes
suppressed summer enhancements to merely 20 % of winter levels despite a 19 % larger summer footprint (Figs. 7 and S3).
The seasonal contrast was secondarily modulated by anthropogenic factors, notably natural gas consumption surges for heating
and holiday cooking during Lunar New Year/New Year festivities. This seasonal trend parallels Los Angeles' fossil fuel minima
in July–September and maxima in December–March (Kim et al., 2025), with analogous cold-season amplification documented
in Indianapolis eddy flux observations (Wu et al., 2022b). Notably, summer $CO_2bio$ at PY exceeded winter levels, aligning
with NDVI seasonality (summer NDVI was 11 % higher than winter) and contrasting temperatures: summer averaged 30.21 °C,
near the optimal canopy temperature for photosynthesis (Liu et al., 2015), while winter was considerably cooler at 17.38 °C
(Table S2). Summer afternoons revealed biogenic fluxes offsetting 60.17 % of anthropogenic emissions—highlighting critical
biogenic modulation of coastal urban carbon budgets. This magnitude resonates with Los Angeles' growing-season biogenic
consumption of 60 % fossil emissions (Kim et al., 2025), while New York City's summer biogenic uptake offsets 40 % of
anthropogenic enhancements, fully neutralizing traffic emissions in highly congested zones (Wei et al., 2022). Crucially, data
from Indianapolis demonstrating equivalent warm-season amplitudes between $CO_2bio$ and $CO_2ff$ underscore the necessity of
accounting for biogenic fluxes in urban $CO_2ff$ quantification (Wu et al., 2022b).

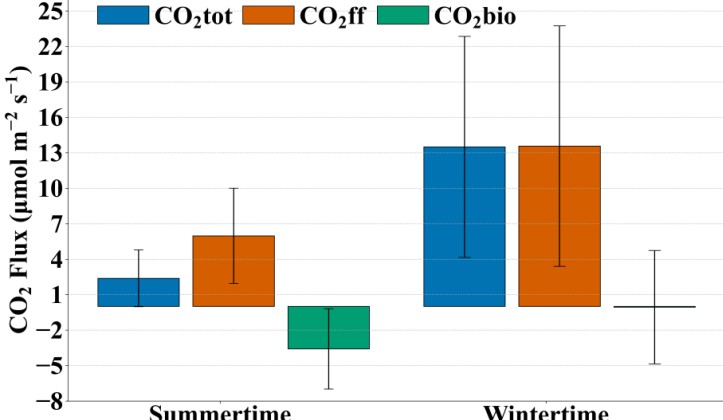


**Figure 9.** Average $CO_2tot$, $CO_2ff$, and $CO_2bio$ derived from $CO_2$ enhancement observations at the PY station, during summer and
winter afternoon hours (12:00–16:00). Error bars indicate ±1 SD.
The maximum winter afternoon $CO_2ff$ at Guangzhou's urban site reached 13.62 ± 9.38 μmole m$^{-2}$ s$^{-1}$. While uncertainties
from in situ observations and background concentrations alone would yield a marginal flux error of 0.36 μmole m$^{-2}$ s$^{-1}$,
atmospheric transport introduces substantial unquantified uncertainty—consistent with documented winter wind speed
overestimations in meteorological models (Yadav et al., 2021) and biases in simulating boundary layer-free troposphere

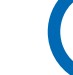

exchange (Lin et al., 2021). Though transport uncertainty quantification falls beyond this study's scope, we identified surface
flux sensitivity zones via winter footprint distributions in Fig. 10 (bounded areas), where influence peaks near receptor sites
(Wu et al., 2022). Temporally resolved EDGAR emissions within these sensitivity zones yielded a bottom-up $CO_2$ff estimate
of 19.81 μmole $m^{-2}$ $s^{-1}$. Our observation-model-$R_{CO}$ derived $CO_2$ff (13.62 μmole $m^{-2}$ $s^{-1}$) showed reasonable convergence
despite a 6.19 μmole $m^{-2}$ $s^{-1}$ discrepancy—primarily attributable to transport uncertainties, paralleling the 3.5 μmole $m^{-2}$ $s^{-1}$
gap in Los Angeles attributed to mixed-layer height and wind speed errors (Kim et al., 2025). Summer afternoons exhibited
peak $CO_2$bio of $-3.67 \pm 3.48$ μmol $m^{-2}$ $s^{-1}$. This aligns with the modeled Pearl River Delta net ecosystem exchange (NEE)
range of $-0.1$ to $-12$ μmol $m^{-2}$ $s^{-1}$ (Mai et al., 2024a), U.S. urban observations such as Los Angeles with $-6.7 \pm 0.7$ μmol $m^{-2}$
$s^{-1}$, and broader reported biogenic flux ranges of 0 to $-15$ μmol $m^{-2}$ $s^{-1}$ (Wu et al., 2021; Wei et al., 2022; Kim et al., 2025).
Crucially, bottom-up inventories proved inherently ill-equipped to resolve summer $CO_2$ff variability due to their statistical
nature—exhibiting reductions in summer versus winter that were significantly smaller than observationally derived
quantifications, at merely a quarter of the latter—and their inability to capture coastal atmospheric dynamics. This fundamental
limitation, therefore, underscores the critical value of real-time monitoring for urban carbon flux assessment.

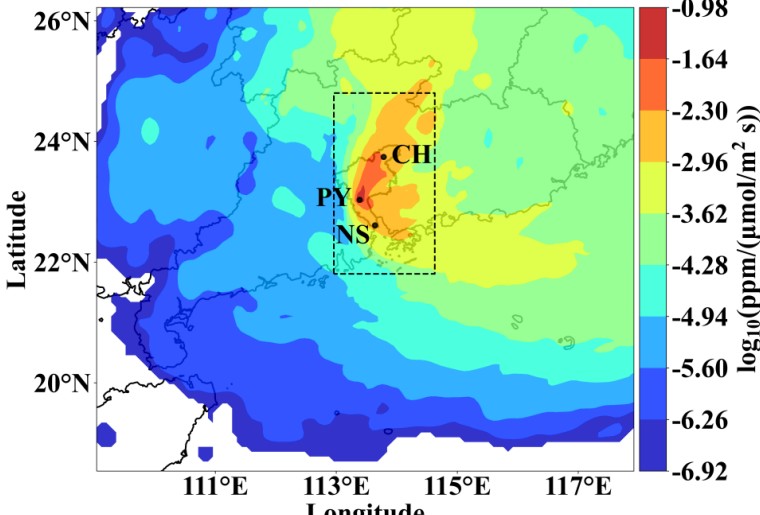


**Figure 10.** Spatial distribution of winter afternoon footprints at the PY station, highlighting high-sensitivity surface flux regions.
**4 Conclusions**
This study develops a novel observation-based framework that integrates high-precision atmospheric $CO_2$ measurements,
meteorological analyses, and $\Delta CO/\Delta CO_2$ ratios (Rco) to disentangle the key drivers of $CO_2$ dynamics in the coastal megacity
of Guangzhou. The analysis, covering the period from January 2023 to September 2024, successfully quantifies the respective
roles of anthropogenic emissions, biogenic fluxes, and meteorological processes. The results demonstrate distinct regional





drivers of $CO_2$ variability: atmospheric transport dominates the large seasonal amplitude (25.63 ppm) at the coastal site,
biogenic fluxes drive substantial diurnal cycles in suburban areas (peak summer amplitude: 39.90 ppm), and anthropogenic
emissions exert primary control in the urban core. Notably, sea-land breeze (SLB), typically regarded as a ventilation
mechanism, are found to amplify afternoon $CO_2$ accumulation by +2.08 ppm during summer under stable boundary layer
conditions. During the peak growing season, urban biogenic fluxes offset 60.17 % of anthropogenic $CO_2$ emissions. Spatially
heterogeneous Rco values, including an urban regression slope of 7.45 ± 1.38 ppb $ppm^{-1}$, confirm trends toward improved
combustion efficiency.

The synthesis of these findings underscores that the $CO_2$ budget of a coastal megacity is governed by complex, non-linear

interactions among emission sources, biospheric activity, and both synoptic and local meteorology. Our framework effectively
partitions anthropogenic and biogenic contributions, revealing that biogenic fluxes are not merely a background signal but a
dynamic and decisive component of the urban carbon cycle. Crucially, coastal meteorology exhibits a dual role: while SLB
often facilitate ventilation, they can also trap and recirculate pollutants, leading to unanticipated $CO_2$ accumulation under
specific atmospheric stratification.

Compared to existing studies, our innovative contributions to the current body of knowledge are as follows: (1) While

prior research has largely overlooked the quantitative assessment of biogenic fluxes and their contribution to urban-scale $CO_2$
budgets, our study demonstrates that biogenic fluxes can offset up to 60.17 % of anthropogenic emissions during the peak
growing season in a coastal megacity. This highlights the critical role of biogenic processes in urban carbon dynamics and
overall $CO_2$ budgeting, underscoring the necessity of accurately accounting for these fluxes in future fossil fuel inversion
studies. This finding helps address the gap in observational constraints and contribution assessments of biogenic fluxes in
urban carbon cycle research. (2) Although the influence of SLB on urban $CO_2$ dynamics remains underexplored in existing
literature, our work reveals a non-linear response of $CO_2$ to SLB, which can exacerbate $CO_2$ accumulation under stable
boundary layer conditions—a critical feedback mechanism previously overlooked in urban carbon assessments. These results
fill a fundamental knowledge gap regarding how $CO_2$ dynamics respond to SLB within coastal urban carbon cycle science. (3)
The summer discrepancies between our observed $CO_2ff$ and the bottom-up inventory suggest that such inventories struggle to
capture the high-frequency variability induced by coastal atmospheric dynamics, an aspect that has been poorly documented
in earlier work.

Despite validation against inventories, the NDVI, and independent studies confirming the robustness of our framework,

several limitations remain. First, the spatial resolution of our three-station network limits the ability to resolve hyperlocal
emission sources, such as individual traffic corridors. Second, gaps in high-resolution meteorological data, particularly in the
suburban area, constrain a more comprehensive analysis of biosphere-weather interactions. Moreover, our identification of
SLB events relies on 2D wind fields and does not incorporate full 3D boundary layer structures; however, the consistency of



our SLB identification with previous studies in the Pearl River Delta urban agglomeration suggests that this limitation does
not significantly affect the core findings. Additionally, the derivation of biogenic fluxes as a residual of the $CO_2$ budget
propagates uncertainties associated with the Rco tracer ratio and neglects minor contributions from secondary CO formation
(< 1 %). Although beyond the scope of this study, uncertainties in atmospheric transport simulations remain a major factor
contributing to flux estimation discrepancies. These limitations outline a clear pathway for future work, including the
deployment of dense low-cost sensor networks, refinement of Rco estimates using VOC-based proxies, development of
uncertainty-quantified coastal boundary layer models, and vegetation-specific flux measurements with isotopic constraints.

We show that coastal meteorology can override fundamental emission patterns, offering new insights into atmospheric

$CO_2$ dynamics and emission inversion in coastal cities. These findings help reduce uncertainties in $CO_2$ inversion estimates
across such regions. At the same time, rising atmospheric $CO_2$ levels—a major driver of global warming—intensify land-sea
thermal contrast along coastlines, leading to more frequent sea-land breeze events. We further reveal that these breezes
significantly influence coastal $CO_2$ dynamics. Consequently, the complex interactions and feedback mechanisms between sea-
land breezes and $CO_2$ merit in-depth investigation in coastal atmospheric science. Moreover, by quantifying the substantial
offset of anthropogenic $CO_2$ emissions by biogenic fluxes, our results underscore the need for high-resolution models of urban
biogenic fluxes. This can be achieved through the installation of additional urban flux towers, improved constraints on $CO_2$
fluxes from urban lawns and mixed vegetation, and enhanced empirical parameterizations for biosphere models in urban
settings. This work provides a scientific basis for urban policymakers to promote integrated gray-green infrastructure and
harness ecological processes for climate mitigation. Finally, the methodology presented here—including the confirmed decline
in urban combustion efficiency in response to policy interventions—can be extended to other cities to evaluate the effectiveness
of climate mitigation strategies.
**Code and data availability.** The STILT model source code used in this paper has been published on Zenodo and can be
accessed at https://doi.org/10.5281/zenodo.1196561 (Fasoli, 2018). The EDGAR data used in this study are publicly available
at https://edgar.jrc.ec.europa.eu/dataset_ghg2024#conditions (last access: 18 June 2025)(Crippa et al., 2024). The planetary
boundary layer height data used in this study are available at https://doi.org/10.24381/cds.adbb2d47 (Hersbach, 2023). The
NDVI data used in this study are available at https://doi.org/10.5067/MODIS/MOD13A3.006 (Didan, 2015). The
CarbonTracker (CT-NRT.v2024-5) products are available online at https://doi.org/10.15138/ATPD-K925 (Jacobson et al.,
2024). The NOAA Earth System Research Laboratory/Global Monitoring Laboratory (NOAA GML) data used in this study
are available at https://doi.org/10.25925/20241101 (Schuldt et al., 2024). Additional data and information used in this study
are available from the corresponding author upon request.
**Author contributions.** JWZ and ML designed the study. JWZ, YJL, CLP, BH, YYH, XFL, SJS, CLC, CW, ZZ, JJL and ML
contributed to data collection and data analysis. JWZ designed and performed the model simulations. JWZ and ML wrote the
paper with contributions from all coauthors. JJL and SJS provided valuable feedback and opinions for paper refinement. All
the authors revised the paper and edited the text.





**Competing interests.** The contact author has declared that none of the authors has any competing interests.
**Disclaimer.** Publisher's note: Publisher's note: Copernicus Publications remains neutral with regard to jurisdictional claims in
published maps and institutional affiliations.
**Acknowledgements.** The authors would like to thank the personnel who participated in data collection, instrument
maintenance, and logistic support during the field campaign. We also acknowledge the NOAA GML for providing the $CO_2$
GLOBALVIEWplus v10.1 ObsPack and CarbonTracker CT-NRT.v2024-5 datasets, which were used for monitoring
comparison in this study. CarbonTracker CT-NRT.v2024-5 results provided by NOAA GML, Boulder, Colorado, USA from
the website at http://carbontracker.noaa.gov.
**Financial support.** This work was financially supported by the National Natural Science Foundation of China (Grant no.
42477273) and the National Key R&D Program of China (Grant no. 2022YFE0209500).

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
