# Peer review of "Atmospheric CO2 dynamics in a coastal megacity: spatiotemporal"

_EGUsphere, 2025_

## Author Comment (AC1)

**Response to RC1:**

This paper focuses on the atmospheric $CO_2$ dynamics in Guangzhou, a coastal megacity. It does this by developing an observation-based framework that integrates ground-based CO and $CO_2$ observations, as well as $\Delta CO/\Delta CO_2$ ratios. The paper analyses the spatiotemporal patterns of $CO_2$, assesses the influence of sea–land breezes and partitions anthropogenic and biogenic emissions. While the study presents interesting findings regarding Guangzhou's carbon emissions, some clarification of the results is necessary. The manuscript could be improved by providing more comprehensive interpretations of the methods and results, particularly with regard to the uncertainty of the results presented. Please see my comments below.

Response:

We appreciate the reviewer for the constructive overview and insightful suggestions, which have been very helpful in improving the quality of the manuscript. Guided by these comments, we have revised the manuscript to improve the clarity and rigor of the methods and results and to better communicate uncertainties and limitations where relevant. Below we provide point-by-point responses and indicate the exact locations where the corresponding revisions were implemented in the revised manuscript. The revised text has been highlighted in the manuscript for ease of reference.

**General Comments:**

(1) CO was measured simultaneously, but is only analysed in Section 3.4. Since CO is a good proxy for $CO_2$ over urban regions, it would be interesting to check the CO results, for example in Figures 4 and 6. Please show and interpret the diurnal cycle of CO in contrast to $CO_2$.

Response:

We agree that CO provides an important combustion tracer to interpret $CO_2$ variability. To directly address this, we added two new supplemental figures and concise cross-species interpretation in the main text. Specifically, we now show (i) synchronous CO diurnal cycles across seasons and weekdays/weekends for NS, PY and CH (new Fig. S9), and (ii) CO diurnal cycles during SLB days versus non-SLB days at NS (new Fig. S11) to complement the $CO_2$ results in Figs. 5 and 7 (formerly Figs. 4 and 6). We kept Figs. 5 and 7 focused on $CO_2$ to avoid overcrowding, but we added explicit caption cross-references to Figs. S9 (Line 438) and S11 (Lines 510–511) and summarized the key contrasts in Sect. 3.1.2 (Lines 464–473) and Sect. 3.2 (Lines 512–520).

In these added texts, we highlight three robust CO–CO₂ contrasts: (1) the morning CO peak aligns with the CO₂ morning maximum at PY (combustion/traffic influence), (2) CO lacks a pronounced mid-afternoon minimum (supporting biogenic control of the CO₂ midday drawdown), and (3) CO confirms the seasonally opposite SLB regimes (ventilation in cooler seasons vs summertime trapping/recirculation). For consistency, we made corresponding brief updates in the Abstract and Conclusions to note this added CO–CO₂ contrast (Lines 23–26; Lines 733–738).

[Figure]

**Figure S9.** Diurnal CO variations at the (a–b) NS, (c–d) PY, and (e–f) CH stations across (A) seasons and (B) weekdays/weekends. Seasons are defined as Spring (Mar–May), Summer (Jun–Aug), Autumn (Sep–Nov), and Winter (Dec–Feb). Error bars indicate ± 1 SD.

[Figure]

**Figure S11.** Diurnal variations in CO concentrations, wind direction, and wind speed at the coastal station (NS) during sea–land breeze days (SLBD) and non-SLB days (NSLBD) by season. Error bars indicate ± 1 SD.

(2) Please explain how the error bars in Figure 9 are calculated. It seems to me that the uncertainty is so large that the difference between summer and winter is not significant. Also, what is the uncertainty of the estimation in line 434 (60.17%)?

Response:

We appreciate the reviewer's question about the error bars in Fig. 10 (formerly Fig. 9) and the uncertainty of the reported summertime offset fraction. Figure 10 summarizes afternoon (12:00–16:00 LT) daily means at PY for July 2023 (summer; n = 29 valid days) and December 2023 (winter; n = 18 valid days). For each valid day, we first computed the 12:00–16:00 mean $CO_2$tot, $CO_2$ff, and $CO_2$bio, and then formed the monthly statistics from these daily afternoon means (bars: monthly mean; error bars: ±1 SD across daily means). Thus, the error bars represent day-to-day atmospheric variability in the daily afternoon means—driven mainly by transport/ventilation—rather than the uncertainty of the monthly mean. December has fewer valid days because incomplete-afternoon days (e.g., instrument

downtime/maintenance) were excluded by objective QC; minor numerical differences from the original submission reflect consistent application of this valid-day criterion and do not affect the conclusions (e.g., December $CO_2$ff mean ± SD: 13.62 ± 9.38 originally vs 13.56 ± 10.17 here).

Because SD reflects variability (not mean uncertainty), overlap of ± 1 SD ranges does not imply that the July–December contrast is insignificant. To directly address significance, we now report the standard error (SE) and 95 % CI of the monthly means and formal comparisons based on daily means (new Table S5). These show that the winter–summer differences remain statistically detectable for $CO_2$tot, $CO_2$ff, and $CO_2$bio (Welch test; p values reported in Table S5), with consistent inferences from Mann–Whitney tests and bootstrap confidence intervals. We also report robust distributional metrics (median and IQR), which corroborate this significant seasonal increase despite partial day-to-day overlap (Table S5).

The "60.17 %" in the original manuscript denotes the summertime biogenic offset fraction, defined as |$CO_2$bio|/$CO_2$ff using July afternoon monthly means. Using the harmonized Fig. 10 values, the offset is |−3.59|/5.97 = 0.6013 (60.13 %). We quantify its uncertainty using bootstrap resampling of paired daily ($CO_2$ff, $CO_2$bio) values (preserving their covariance), yielding 60.13 % with a 95 % CI of 48–72 %.

These clarifications and updates have been implemented in the revised Fig. 10 caption (Lines 635–640) and in Sect. 3.5 (Lines 610–633). The significance testing, SE, and 95 % confidence intervals for the July–December contrasts are summarized in Table S5, while the summertime offset uncertainty (bootstrap 95 % CI of 48–72 %) is reported in Sect. 3.5 (Lines 705–708). For completeness, we briefly note these uncertainty updates in the Conclusions (Lines 746–749).

[Figure]

**Figure 10.** Average afternoon (12:00–16:00 LT) $CO_2$tot, $CO_2$ff, and $CO_2$bio at PY for July 2023 (summer; n = 29 valid days) and December 2023 (winter; n = 18 valid days). December has fewer valid days because objective QC excluded days with incomplete afternoon coverage (e.g., instrument downtime/maintenance), so the smaller winter

sample reflects data availability rather than subjective selection. Bars show monthly means of daily afternoon values. Error bars indicate ± 1 standard deviation (SD) across daily afternoon means within each month (day-to-day atmospheric variability), not the standard error (SE) of the monthly mean; SE and confidence intervals are reported in Table S5.

**Table S5.** July–December contrasts in daily afternoon means (PY, 12:00–16:00 LT).
($\Delta$ = winter − summer; units: $\mu$mol m$^{-2}$ s$^{-1}$)

| Component | July Mean ± SE | Dec Mean ± SE | July → Dec (median, IQR) | $\Delta$ (Welch 95 % CI) | Welch p | Mann–Whitney p | Bootstrap 95 % CI of $\Delta$ |
|---|---|---|---|---|---|---|---|
| CO$_2$tot | 2.38 ± 0.45 | 13.50 ± 2.20 | 2.00 (0.72–3.05) → 10.34 (7.16–15.00) | +11.12 [6.40, 15.83] | 9.83× 10$^{-5}$ | 4.06×10$^{-7}$ | [7.10, 15.67] |
| CO$_2$ff | 5.97 ± 0.75 | 13.56 ± 2.40 | 4.33 (3.58–7.81) → 10.70 (6.74–18.28) | +7.59 [2.36, 12.82] | 0.0066 | 0.002 | [3.09, 12.65] |
| CO$_2$bio | −3.59 ± 0.63 | −0.06 ± 1.13 | −3.17 (−5.42– −1.39) → −0.20 (−2.11–2.59) | +3.53 [0.87, 6.19] | 0.011 | 0.003 | [1.01, 6.04] |

(3) The manuscript identifies sea–land breeze (SLB) days based on 24-hour wind direction transitions and a wind speed threshold of <10 m s$^{-1}$. While this threshold excludes most strong winds, the authors should clarify whether the potential influence of tropical cyclones or their peripheral circulations was considered. Even when wind speeds remain below 10 m s$^{-1}$, such events can disrupt local wind directions, potentially disturbing the regular daytime–night-time SLB pattern. Without addressing these effects, SLB identification and subsequent CO$_2$ dynamics interpretation may be biased.

Response:

We appreciate the reviewer's careful comment and agree that tropical cyclones (TCs) or their peripheral circulations could disrupt local wind-direction reversals even under moderate winds. We have implemented these clarifications in Sect. 2.4 (Lines 218–241), where we also note that excluding the single overlapping day does not alter our SLB–CO$_2$ interpretation, and we summarize the TC screening in Table S2 (new). We address this in three ways:

**1) Clarifying the wind-speed threshold (wording vs. implementation).** We agree that our original wording "daily mean wind speed < 10 m s$^{-1}$" could be interpreted ambiguously. We have revised Sect. 2.4 to state explicitly that the <10 m s$^{-1}$ criterion is applied to hourly winds over the entire candidate SLB day (00:00–23:00 LT), i.e., no hourly wind-speed value exceeds 10 m s$^{-1}$. Importantly, this hourly cap was already used in our original SLB-day classification; the revision corrects the description for

transparency. Reapplying the clarified criteria reproduces the same SLB-day calendar (number and dates unchanged).

**2) Two-phase (night/day) requirement and local-wind direction.** Our SLB definition requires both a night-time land-breeze phase (01:00–09:00 LT; 302–45°) and a daytime sea-breeze phase (12:00–20:00 LT; 112–202°), each persisting for ≥4 h (or ≥4 h within any running 5 h window). Directional persistence is evaluated using the local-wind direction (after removing the daily-mean background vector), while the wind-speed screen is applied to the observed wind-speed magnitude at 48 m. Together, these requirements reduce the likelihood of misclassifying strongly forced days, because synoptic/TC-peripheral regimes often impose a prolonged anomalous wind pattern rather than a clean diurnal reversal (Atkins and Wakimoto, 1997; Allouche et al., 2023).

**3) Targeted TC screening.** To explicitly assess residual TC contamination, we cross-referenced the SLB-day calendar against 2023 Pearl River Delta (PRD)/Guangzhou TC impact windows compiled from the official Guangdong–Hong Kong–Macao Greater Bay Area (GBA) Climate Monitoring Bulletin (new Table S2). For each TC, Impact Start/End are defined as the first/last local dates on which the bulletin reports PRD/Guangzhou impacts or advisories attributable to that system (including peripheral rainbands/gusts). Because the bulletin is date-based, we conservatively treat the entire day within each window as potentially TC-influenced. Only one SLB day (2 Sep 2023) overlaps these windows; excluding it leaves results unchanged.

Overall, these revisions clarify the wind-speed screening implementation and explicitly assess potential TC/peripheral influences on SLB identification, indicating no bias in the identified SLB-day set or the associated $CO_2$ interpretation.

Atkins, N. T. and Wakimoto, R. M.: Influence of the synoptic-scale flow on sea breezes observed during CaPE, Monthly weather review, 125, 2112-2130, https://doi.org/10.1175/1520-0493(1997)125<2112:IOTSSF>2.0.CO;2, 1997.

Allouche, M., Bou-Zeid, E., and Iipponen, J.: The influence of synoptic wind on land–sea breezes, Quarterly Journal of the Royal Meteorological Society, 149, 3198-3219, https://doi.org/10.1002/qj.4552, 2023.

**Table S2.** Summary of 2023 tropical cyclones (TCs) and impact windows in the Pearl River Delta (PRD)

| TC Number | International Name | Impact Start | Impact End | PRD / Guangzhou Impacts (summary) | Primary Reference |
|---|---|---|---|---|---|
| 2304 | Talim | 2023-07-15 | 2023-07-18 | Gusty winds and storm surge along the western PRD; strong winds and squally showers in Guangzhou on Jul 17. | 2023 Guangdong–Hong Kong–Macao Greater Bay Area (GBA) Climate Monitoring Bulletin (https://my.weather.gov.hk/en/wxinfo/pastwx/2023/files/GD_HK_Mac_GBA_2023.pdf) |
| 2305 | Doksuri | 2023-07-24 | 2023-07-29 | Peripheral rainbands and thunderstorms; local advisories issued in PRD/Guangzhou. | |
| 2309 | Saola | 2023-09-01 | 2023-09-03 | Severe gales and heavy rain across PRD; service suspensions; multiple warnings in Guangzhou and surrounding cities. | |
| 2311 | Haikui | 2023-09-05 | 2023-09-11 | Remnant low brought prolonged heavy rain in PRD; locally record-breaking September rainfall in parts of Guangdong. | |
| 2314 | Koinu | 2023-10-05 | 2023-10-09 | Sustained gales and heavy rain in PRD; transport/service disruptions; multiple warnings in Guangzhou. | |
| 2316 | Sanba | 2023-10-19 | 2023-10-20 | Peripheral effects in PRD with rain/gusts; main impacts over western Guangdong (Zhanjiang/Maoming) and Hainan. | |

**Other Comments:**

Although the introduction highlights three key knowledge gaps in the existing research, it does not emphasise their relevance to coastal cities enough, nor does it distinguish these gaps from those in studies of inland cities. Further descriptions are required. While the introduction mentions Guangzhou's GDP, population, green coverage and sea–land breeze frequency, it does not link these to the study objectives. Green coverage, which is important for biogenic fluxes, is neither compared with that in other coastal cities nor discussed in terms of its impact on flux magnitude. The frequency of the sea–land breeze is cited without detailing its seasonal patterns or how it differs from that in other cities. Furthermore, Guangzhou's carbon mitigation policies, which could influence anthropogenic and biogenic $CO_2$ emissions, are not mentioned.

Response:

    Thanks for the constructive and detailed comments. We have revised and partially restructured the Introduction to strengthen the coastal relevance of the three knowledge gaps, explicitly distinguish coastal from inland regimes, and link Guangzhou's city characteristics and policy context directly to our study

objectives.

**1) Coastal relevance and coastal-inland distinction.** We added explicit language that coastal megacities face attribution challenges driven by land–sea contrasts, marine background inflow, and diurnal reversals in advection/boundary-layer structure, and we contrast this with typical inland regimes lacking marine inflow–outflow (Introduction; Lines 46–52; Lines 76–81).

**2) SLB frequency: seasonal patterns and cross-region contrasts.** We expanded the SLB description to include its seasonal dependence in the Pearl River Estuary. We also added a concise comparison with other Chinese coastal regions (e.g., the Bohai Rim and Yangtze River Delta) to underscore regional heterogeneity in mesoscale transport and background-wind control (Introduction; Lines 74–76; Lines 81–84).

**3) Green coverage: cross-city comparison and implication for flux magnitude.** We added a short cross-city comparison of reported built-up-area green coverage among major Chinese coastal megacities (e.g., Guangzhou, Shenzhen, Qingdao, and Tianjin) and explicitly state the implication that biogenic exchange can be non-negligible when interpreting urban observations, with brief supporting examples from the literature (Introduction; Lines 95–105).

**4) Explicitly framing the three gaps for coastal settings and distinguishing from inland regimes.** We rewrote the three "knowledge gaps" to be explicitly coastal-focused (including SLB-driven transport/mixing and representativeness issues) and to distinguish them from inland-city regimes (Introduction; Lines 107–112). We restate these coastal-focused gaps at the start of the Conclusions for continuity (Lines 717–718).

**5) Linking Guangzhou indicators and policy context to the objectives.** We revised the Guangzhou context so that GDP/population, greening, and frequent SLB are no longer purely descriptive; instead, together with a brief mitigation-policy framing (e.g., Guangdong ETS and key sector measures), they are explicitly tied to the three coastal-focused gaps and used to motivate our objectives and observation-driven framework for robust source–sink attribution in a policy-relevant coastal megacity setting (Introduction; Lines 112–139). To reflect this framing, we updated the Abstract and Conclusions to more explicitly state the mitigation relevance (Lines 29–31; Lines 761–770).

Line 25: clean air policies? Please provide a little bit more details of these policies in the abstract.
Response:

Thanks for the comment. We revised the Abstract and expanded the policy context in the main text

(Sect. 3.4) to make the "clean-air policies" reference more concrete while keeping claims proportionate to our evidence and within the Abstract word limit.

**1) Abstract update.** We replaced the generic phrase with a compact sector-level description, stating that the regression-derived $R_{CO}$ is consistent with the reported post-2013 tightening of coal/industrial and vehicle-emission controls (Lines 26–27). For consistency, we made a matching wording edit in the Conclusions (Lines 725–727).

**2) Main-text expansion.** We added specific programmes and sector measures (coal/power and vehicle controls, including ULE retrofits and China 6 standards) and supporting context, and we explicitly frame this as a consistency check rather than causal attribution, noting other possible contributors (fuel mix, fleet composition, atmospheric oxidation) (Sect. 3.4; Lines 572–581; Lines 587–588).

Line 114: EDGAR full name.

Response:

Thanks for pointing out our mistake. We have revised this in Lines 158–159.

Figure 8: Please show the density plot; the points may overlap strongly with each other.

Response:

Thanks for the suggestion. To reduce overplotting and better visualize the distribution, we revised Fig. 9 (formerly Fig. 8) by replacing the scatter with a 2D histogram density plot (hist2d; 200 × 200 bins), with color indicating the number of paired observations per bin (see colorbar). This highlights the high-density core and sparse tails while retaining the full dataset. Regression lines and reported slope/correlation statistics are computed from the underlying paired data and are unchanged. We updated Fig. 9 and its caption accordingly (Lines 565–569).

[Figure]

**Figure 9.** Seasonal relationships between $\Delta CO_2$ and $\Delta CO$ enhancements at the (a–c) NS, (d–f) PY, and (g–i) CH stations, analyzed using geometric-mean regression. Panels are shown as 2D histogram density plots (hist2d; $200 \times 200$ bins), where color indicates the number of paired observations per bin. The fitted slope represents the $\Delta CO/\Delta CO_2$ emission ratio ($R_{CO}$; ppb ppm$^{-1}$), reported as mean $\pm$ 1 SD (reflecting temporal variability).

Line 448: is the EDGAR mean yearly or monthly? How is the temporal resolution of EDGAR done? How are daytime and night-time emissions differentiated?

Response:

Thanks for your careful comment. The EDGAR product used here is the annual 2023 gridded inventory at $0.1° \times 0.1°$ (EDGAR_2024_GHG; Crippa et al., 2024). To obtain sub-daily variability and distinguish daytime versus nighttime emissions, we temporally disaggregated the annual totals to an hourly series using the high-resolution temporal profiles of Crippa et al. (2020). Daytime/nighttime (and winter-afternoon 12:00–16:00 LT) values are computed by selecting and averaging the corresponding local-time hourly bins. We describe this processing and report the resulting winter-afternoon benchmark over the footprint-defined sensitivity region (Fig. 12) in Sect. 3.5 (Lines 672–675).

Crippa, M., Solazzo, E., Huang, G., Guizzardi, D., Koffi, E., Muntean, M., Schieberle, C., Friedrich, R., and Janssens-Maenhout, G.: High resolution temporal profiles in the Emissions Database for Global Atmospheric Research, Scientific Data, 7, https://doi.org/10.1038/s41597-020-0462-2, 2020.

The STILT model releases 500 particles with a 72-hour backward trajectory and a spatial resolution of $0.08° \times 0.08°$, but no sensitivity tests are reported. It is unclear whether increasing the number of particles or improving the spatial resolution would significantly affect the footprint simulations. These model parameters directly impact the accuracy of $CO_2$ emission estimates, so relevant validation analyses are essential.

Response:

Thanks for highlighting the need to document sensitivity to STILT parameters. We agree and have added a targeted winter paired-day sensitivity analysis at PY to quantify how STILT setup choices affect inferred fluxes. Starting from the baseline configuration (500 particles, 0.08° grid, 72 h backward), we independently varied (1) particle number (1000, 2000), (2) grid resolution (0.05°, 0.10°), and (3) backward duration (96 h, 120 h). For each variant we recomputed footprints, reran the flux-estimation workflow, and compared against the baseline using paired daily afternoon means (12:00–16:00 LT; n = 18) with effect sizes (percent/absolute differences), Pearson r, and 95 % CIs (with paired t-tests reported only as detectability indicators at this sample size).

These additions are described in the Methods in Sect. 2.5.1 (Lines 294–302) and Sect. 2.5.2 (Lines 333–335), and the quantitative outcomes are reported in Sect. 3.5 (Lines 641–664) and summarized in Fig. 11 (new) and Tables S6–S7 (new).

The results show that inferred winter-afternoon fluxes at PY are robust to these STILT setup choices: increasing particle number to 1000/2000 changes $CO_2$ff by −0.56 %/−0.24 % and $CO_2$tot by −0.52 %/−0.31 %; refining the grid to 0.05° yields similarly small decreases ($CO_2$ff: −0.72 %; $CO_2$tot: −0.78 %); and extending the backward duration to 96/120 h produces changes of −1.34 %/−1.05 % ($CO_2$ff) and −1.33 %/−1.31 % ($CO_2$tot). Only the intentionally coarser 0.10° grid produces a small but detectable decrease ($CO_2$ff: −1.47 %; $CO_2$tot: −1.40 %), while all other settings yield changes ≤1.34 % with 95 % CIs spanning zero. Day-to-day consistency remains essentially unchanged (r ≈ 0.999; RMSE 0.28–0.45 $\mu$mol m$^{-2}$ s$^{-1}$; Fig. 11; Table S6). $CO_2$bio shows similarly robust behavior: because wintertime $CO_2$bio is near zero at PY, we assess it in absolute terms, and the test–baseline differences remain small with 95 % CIs generally spanning zero (Table S7). The across-run day-by-day ensemble spread is also tightly

bounded (median 0.20–0.21 μmol m$^{-2}$ s$^{-1}$; median CV ≈ 1.8 %), and paired-day scatter remains close to 1:1. Overall, these results indicate that our baseline STILT configuration is in a converged regime and that the inferred winter CO$_2$ff dominance is robust to reasonable transport-parameter choices. We also made corresponding wording updates in the Abstract and Conclusions to reflect this robustness check (Lines 27–28; Lines 743–746).

[Figure]

**Figure 11.** STILT parameter sensitivity at PY (winter). Panel A: mean percent difference (variant − baseline) of inferred fluxes relative to the winter baseline (500 particles, 0.08°, 72 h), computed from paired daily afternoon means (12:00–16:00 LT; n = 18); Δ% = (variant − base)/base × 100; negative values indicate lower than baseline. Panel B: paired scatter of CO$_2$ff (μmol m$^{-2}$ s$^{-1}$) from each variant versus the baseline for the same days; solid line is 1:1 (y = x).

**Table S6.** Wintertime (12:00–16:00 LT) paired-day sensitivity of PY inferred fluxes to STILT parameter choices (n = 18). Variants (particle number, grid spacing, backward duration) are compared with the baseline (500 particles, 72 h, 0.08°). Metrics report effect size (pct_diff_% and 95 % CI), day-to-day consistency (Pearson r), RMSE (μmol m$^{-2}$ s$^{-1}$), and detectability (paired t-test p value). Upper block: CO$_2$ff; lower block: CO$_2$tot. Across the baseline plus six variants, the day-by-day ensemble spread—computed as the standard deviation across the seven runs for each day and then summarized by the median—was 0.20–0.21 μmol m$^{-2}$ s$^{-1}$ (median CV ≈ 1.8%).

| metric | comparison | pct_diff_% | pearson_r | rmse | p_value | ci95_lo | ci95_hi |
|---|---|---|---|---|---|---|---|
| | Back 120 h vs Base (72 h) | -1.05 | 0.9994 | 0.38 | 0.1136 | -0.3216 | 0.0376 |
| PY_CO$_2$ff | Back 96 h vs Base (72 h) | -1.34 | 0.9993 | 0.45 | 0.0831 | -0.3904 | 0.0265 |
| | Particles 1000 vs Base (500) | -0.56 | 0.9996 | 0.33 | 0.3410 | -0.2380 | 0.0871 |
| | Particles 2000 vs Base (500) | -0.24 | 0.9995 | 0.31 | 0.6738 | -0.1916 | 0.1269 |

| | | | | | | |
|---|---|---|---|---|---|---|
| | Res 0.05° vs Base (0.08°) | -0.72 | 0.9996 | 0.31 | 0.1917 | -0.2506 | 0.0542 |
| | Res 0.10° vs Base (0.08°) | -1.47 | 0.9997 | 0.39 | 0.0269 | -0.3729 | -0.0257 |
| PY_CO$_2$tot | Back 120 h vs Base (72 h) | -1.31 | 0.9992 | 0.44 | 0.0862 | -0.3819 | 0.0280 |
| | Back 96 h vs Base (72 h) | -1.33 | 0.9992 | 0.45 | 0.0916 | -0.3900 | 0.0322 |
| | Particles 1000 vs Base (500) | -0.52 | 0.9996 | 0.28 | 0.2983 | -0.2098 | 0.0683 |
| | Particles 2000 vs Base (500) | -0.31 | 0.9995 | 0.29 | 0.5502 | -0.1864 | 0.1029 |
| | Res 0.05° vs Base (0.08°) | -0.78 | 0.9996 | 0.31 | 0.1527 | -0.2541 | 0.0431 |
| | Res 0.10° vs Base (0.08°) | -1.40 | 0.9997 | 0.35 | 0.0164 | -0.3394 | -0.0393 |

**Table S7.** Wintertime (12:00–16:00 LT) paired-day sensitivity of PY CO$_2$bio inferred fluxes to STILT parameter choices (n = 18). Variants (particle number, grid spacing, backward duration) are compared with the baseline (500 particles, 72 h, 0.08°). We report the absolute paired-day test–baseline difference, defined as ΔCO$_2$bio = CO$_2$bio(variant) − CO$_2$bio(baseline), summarized by the paired-day mean (Δ; μmol m$^{-2}$ s$^{-1}$) and its 95% confidence interval (CI95_lo, CI95_hi; μmol m$^{-2}$ s$^{-1}$). Because wintertime CO$_2$bio at PY is close to zero, percent differences are not shown. Across-run daily spread of CO$_2$bio—defined as the day-by-day standard deviation across the baseline and all variants—has median 0.045 and IQR 0.016–0.067 μmol m$^{-2}$ s$^{-1}$.

| metric | comparison | mean (ΔCO$_2$bio) | ci95_lo | ci95_hi |
|---|---|---|---|---|
| | Back 120 h vs Base (72 h) | −0.035 | −0.094 | 0.024 |
| | Back 96 h vs Base (72 h) | 0.003 | −0.057 | 0.063 |
| PY_CO$_2$bio | Particles 1000 vs Base (500) | 0.005 | −0.031 | 0.041 |
| | Particles 2000 vs Base (500) | −0.009 | −0.052 | 0.033 |
| | Res 0.05° vs Base (0.08°) | −0.007 | −0.047 | 0.033 |
| | Res 0.10° vs Base (0.08°) | 0.010 | −0.043 | 0.063 |

Figure 2: The distinction between the different wind directions is unclear. The authors should consider optimising the figure, for example by using more distinct colours, line styles or annotations, to improve clarity and readability.

Response:

Thanks for the comment. We agree and have revised Fig. 2 to improve separability among winddirection sectors and overall readability. Specifically, we (i) adopted a high-contrast, color-blind–friendly palette (Okabe–Ito) for the five sectors (Local/NE/SE/SW/NW), (ii) enhanced marker distinguishability by using uniform symbols with clear edges (white outlines and slight transparency to reduce overplotting), and (iii) moved the legend outside the panels and explicitly labeled the sector angle ranges. These changes substantially improve clarity compared with the previous version. We updated Fig. 2 and its caption accordingly (Lines 349–353).

[Figure]

**Figure 2.** Time series of atmospheric $CO_2$ concentrations at the (a) NS, (b) PY, and (c) CH stations. For NS and PY, points are color-coded by wind category: local conditions (wind speed < 1.5 m s$^{-1}$) and four directional sectors for winds ≥ 1.5 m s$^{-1}$ (NE, 0–90°; SE, 90–180°; SW, 180–270°; NW, 270–360°). For CH, wind-direction classification is not shown and the time series is plotted without sector coloring.

Figure 7: It is recommended that the legend be placed outside the figure.

Response:

Thanks for the comment. We have revised Fig. 8 (formerly Fig. 7) by moving the legend outside the figure, as suggested. The time period labels are now placed at the top of the plot for better clarity. We updated Fig. 8 and its caption accordingly (Lines 542–545).

[Figure]

**Figure 8.** Distributions of hourly $CO_2$ enhancement ($\Delta CO_2$) above the marine background at NS, PY, and CH during the (a) overall, (b) summer, and (c) winter periods, shown for all hours, midday (12:00–16:00 LT), and midnight (00:00–05:00 LT). White dots denote the mean values, and white horizontal lines denote the median values.

Line 195: $\Delta CO_2$ appears improperly formatted.

Response:

Thanks for pointing out our mistake. We have revised this in Line 265.

On line 288, 'Despite CH's stronger biogenic coupling (NDVI correlation: −0.72; Fig. 3f), NS's $CO_2$ levels remained 9.80 ppm lower than CH in summer and 5.80 ppm higher in winter, underscoring transport-dominated over biogenic controls at the coastal site.' This whole sentence should be rephrased to enhance

logical rigour.

Response:

Thanks for the comment. We agree that the original wording could be read as conflating CH's strong biogenic coupling with the NS–CH seasonal contrast. We rephrased the sentence to make the logic explicit: CH shows strong biogenic coupling (NDVI–$CO_2$ correlation −0.72; Fig. 4f), whereas the seasonal sign reversal of NS relative to CH (−9.80 ppm in summer; +5.80 ppm in winter) is more consistent with transport and boundary-layer influences at the coastal site. We support this interpretation by referencing NS's weak NDVI coupling (r = −0.08), narrow NDVI range (0.22–0.42), seasonal shifts in marine–continental transport (summer dilution vs. winter urban outflow) and seasonal boundary-layer depth changes (Figs. 2 and S6). These edits are implemented in Sect. 3.1.1 (Lines 412–417).

On line 324, it states that, at CH, 'smaller daytime weekday–weekend differences suggest that biogenic fluxes outweigh anthropogenic variations'. This statement is somewhat too absolute. It would be better to acknowledge the uncertainties more appropriately and consider the potential influence of atmospheric transport and boundary layer dynamics.

Response:

Thanks for the comment. We agree that the original wording was too definitive. We revised Sect. 3.1.2 (Lines 455–458) to soften the inference and to emphasize that a small weekday–weekend contrast at CH is not a unique indicator of source dominance, because transport and boundary-layer mixing can dilute or mask anthropogenic weekday–weekend signals. We therefore frame this result as a qualitative, supportive observation rather than a standalone attribution diagnostic. For consistency, we also made parallel wording edits for PY and NS to keep the interpretation cautious and process-based across sites (Lines 458–462).

---

## Author Comment (AC2)

**Response to RC2:**

This study analyzes $CO_2$ and CO concentration variations over 1 year and 9 months at three sites in Guangzhou—a coastal megacity—examining their relationship with land–sea breezes. Using backward trajectory footprint modeling, it quantifies fossil fuel and biogenic contributions. Given the scarcity of direct $CO_2$ observations in this region, these findings offer valuable insights into carbon sources/sinks in coastal southern Chinese megacities. The methodology is overall sound, and the work merits publication in Atmospheric Chemistry and Physics. However, the introduction of research background, uncertainty analysis in source partition, the robustness of the results, and the depth of discussion could be further improved:

Response:

We thank the reviewer for the thorough and constructive assessment of our manuscript. We are grateful for the reviewer's supportive comments and the encouraging recommendation for publication. We have carefully considered all suggestions and revised the manuscript accordingly to address the main concerns raised. In particular, we have:

1) strengthened the introduction and research motivation;

2) clarified and expanded the treatment of uncertainty in the source-partitioning analysis;

3) improved the robustness presentation and deepened the interpretation and discussion of the results.

We believe these revisions have substantially improved the clarity, rigor, and overall quality of the manuscript. Below we provide point-by-point responses and indicate the exact locations where the corresponding revisions were implemented in the revised manuscript. The revised text has been highlighted in the manuscript for ease of reference.

**Specific comments:**

Introduction Section: The introduction could benefit from restructuring to enhance its logical flow. The rationale for reporting urban-scale $CO_2$ concentrations—particularly the need to clarify carbon sources and sinks—should be more explicitly emphasized. While the study relies heavily on the $CO_2/CO$ ratio and footprint modeling, there is limited discussion on the inherent uncertainties of these methods or how they compare with alternative approaches, such as 3-D atmospheric inversions. Introducing these

methodological considerations would strengthen the foundation for the work.

Response:

We appreciate the reviewer's helpful suggestion. In the revised Introduction, we implemented a targeted restructuring to improve the logical flow by (i) strengthening the rationale for using urban-scale atmospheric $CO_2$ concentrations while emphasizing the source–sink attribution challenge in coastal megacities, (ii) adding a concise methodological context (including 3-D inversions) to clarify how our approach complements existing methods while acknowledging their respective sensitivities/limitations, (iii) explicitly stating the inherent uncertainties of the $CO_2/CO$ ($R_{CO}$) approach and footprint-informed transport, and (iv) adjusting the placement of our framework so that the "gap–site–objectives" narrative is more cohesive.

**1) Clearer rationale for reporting urban-scale $CO_2$ concentrations and need for source–sink clarification.** We added a clearer motivation that concentration observations provide an integrated constraint (anthropogenic emissions + biogenic exchange + transport) but inherently introduce attribution ambiguity. We highlighted why this is especially acute in coastal settings due to marine background inflow and diurnal transport/boundary-layer reversals (Introduction; Lines 40–52). For consistency, we also adjusted the Abstract opening to foreground the mitigation relevance and the coastal attribution challenge (Lines 15–17).

**2) Comparison with alternative approaches, including 3-D inversions.** We added a structured paragraph describing how 3-D inversions can estimate city-scale emissions, while acknowledging practical limitations such as sub-grid representativeness errors and sensitivities to biogenic priors. Accordingly, we position our observation-driven framework as a process-level complement to these inversion approaches (Introduction; Lines 54–71; Lines 119–125). This complementarity is reiterated in the Conclusions (Implications): while our framework does not produce posterior flux fields as in a formal Bayesian inversion, it provides an observation-driven tool for rapid process attribution and consistency checking in coastal urban carbon monitoring and mitigation assessment (Lines 770–773).

**3) Explicit uncertainties in $CO_2/CO$ ($R_{CO}$) and footprint modeling.** We explicitly acknowledged that CO-based inference carries uncertainties (e.g., variable $R_{CO}$ and chemical processing) and that footprint estimates are sensitive to meteorological forcing. we also clarified our evaluation strategy (Introduction; Lines 125–132). These limitations are further described in the Methods uncertainty section (Sect. 2.5.2; Lines 312–315) and summarized in the Conclusions (Lines 756–757).

**4) Improved narrative flow.** To enhance logical progression, we repositioned the description of our framework to follow the explicit gap statements and the justification of Guangzhou as a "living laboratory". This ensures the "solution" is introduced only after the challenges and site context are established (Introduction; Lines 107–132).

Section 2.5: The a priori emission inventory utilized in the study should be better specified, which inventory? What spatial resolution for what year?

Response:

Thank you for the comment. We agree that the wording in Sect. 2.5 could be read as implying that an a priori emissions inventory is required. We therefore revised Sect. 2.5 to state explicitly that our observation-driven framework does not assimilate or rely on any a priori inventory to derive $CO_2tot/CO_2ff/CO_2bio$; it uses only concentration enhancements at the receptor sites together with transport-model footprints (Sect. 2.5; Lines 246–253). We also clarify that emission inventories are mentioned only for contextual description (Sect. 2.1) and for an independent bottom-up comparison (Sect. 3.5), not as priors or constraints in the partitioning.

Because inventories are cited elsewhere in the manuscript, we now explicitly specify the requested details—dataset name, year, and spatial resolution—at the points where each inventory is introduced: the contextual description in Sect. 2.1 (Lines 158–159) and the independent bottom-up comparison in Sect. 3.5 (Lines 672–676). In addition, to ensure consistent messaging and avoid any remaining ambiguity, we made brief alignment edits in the Abstract (Lines 17–20) and Conclusions (Lines 718–721) noting that the partitioning is performed without assimilating emission inventories.

Line 219: for footprint simulation, using 500 particles over a 72-hour simulation period appears rather small. In this setting, the particle count per time step and per grid point is extremely low, potentially introducing substantial uncertainty in footprint estimation. It would be valuable to test and report the sensitivity of results to a larger number of particles (1000-10000) to ensure robustness.

Response:

Thank you for highlighting the need to document sensitivity to STILT particle number. We agree that particle number can influence footprint estimates and thus inferred fluxes. In response, we added a targeted winter-afternoon sensitivity analysis at PY to assess robustness to STILT setup choices. Starting

from our baseline configuration (500 particles, 0.08° grid, 72 h backward), we increased particle number to 1000 and 2000 particles and, in parallel, tested two additional key settings (grid spacing: 0.05° and 0.10°; backward duration: 96 h and 120 h). For each variant, we recomputed footprints, reran the full flux-estimation workflow, and compared $CO_2tot$/$CO_2ff$/$CO_2bio$ against the baseline using paired daily afternoon means (12:00–16:00 LT), reporting effect sizes and 95 % confidence intervals (CIs) as primary quantities.

The test design is described in Sect. 2.5.1 (Lines 294–302), the statistical treatment in Sect. 2.5.2 (Lines 333–335), and results in Sect. 3.5 (Lines 641–664) and summarized in Fig. 11 (new) and Tables S6–S7 (new).

The results show that inferred winter fluxes are robust to increased particle number. Relative to the baseline, increasing particles to 1000/2000 changes $CO_2ff$ by −0.56 %/−0.24 % and $CO_2tot$ by −0.52 %/−0.31 % (paired daily afternoons), indicating near-converged behavior for winter afternoons. This is supported by extremely high baseline–variant correlations ($r \geq 0.999$) and small RMSE values (0.28–0.45 $\mu mol\ m^{-2}\ s^{-1}$; Fig. 11; Table S6). $CO_2bio$ shows similarly robust behavior: because wintertime $CO_2bio$ is near zero at PY, we assess it in absolute terms, and the test–baseline differences remain small with 95 % CIs generally spanning zero (Table S7). Across all tested settings, changes remain small; only the intentionally coarser 0.10° grid yields a statistically detectable but still minor decrease ($CO_2ff$: −1.47 %, $p = 0.0269$; $CO_2tot$: −1.40 %, $p = 0.0164$), while all other variants show changes ≤1.34 % with 95 % CIs spanning zero ($p \geq 0.083$). The across-run day-by-day ensemble spread is also tightly bounded (median 0.20–0.21 $\mu mol\ m^{-2}\ s^{-1}$; median CV ≈ 1.8 %), and paired-day scatter remains close to 1:1. Overall, these additions indicate that the inferred winter fluxes are not artifacts of particle number or reasonable transport-parameter choices. Given the near-converged behavior at 1000–2000 particles, extending to much larger particle numbers is unlikely to materially change the inferred fluxes within our uncertainty context and would substantially increase computational cost. We also made corresponding wording updates in the Abstract and Conclusions to reflect this robustness check (Lines 27–28; Lines 743–746).

[Figure]

**Figure 11.** STILT parameter sensitivity at PY (winter). Panel A: mean percent difference (variant − baseline) of inferred fluxes relative to the winter baseline (500 particles, 0.08°, 72 h), computed from paired daily afternoon means (12:00–16:00 LT; n = 18); Δ% = (variant − base)/base × 100; negative values indicate lower than baseline. Panel B: paired scatter of $CO_2ff$ (μmol m$^{-2}$ s$^{-1}$) from each variant versus the baseline for the same days; solid line is 1:1 (y = x).

**Table S6.** Wintertime (12:00–16:00 LT) paired-day sensitivity of PY inferred fluxes to STILT parameter choices (n = 18). Variants (particle number, grid spacing, backward duration) are compared with the baseline (500 particles, 72 h, 0.08°). Metrics report effect size (pct_diff_% and 95 % CI), day-to-day consistency (Pearson r), RMSE (μmol m$^{-2}$ s$^{-1}$), and detectability (paired t-test p value). Upper block: $CO_2ff$; lower block: $CO_2tot$. Across the baseline plus six variants, the day-by-day ensemble spread—computed as the standard deviation across the seven runs for each day and then summarized by the median—was 0.20–0.21 μmol m$^{-2}$ s$^{-1}$ (median CV ≈ 1.8%).

| metric | comparison | pct_diff_% | pearson_r | rmse | p_value | ci95_lo | ci95_hi |
|---|---|---|---|---|---|---|---|
| PY_CO₂ff | Back 120 h vs Base (72 h) | -1.05 | 0.9994 | 0.38 | 0.1136 | -0.3216 | 0.0376 |
| | Back 96 h vs Base (72 h) | -1.34 | 0.9993 | 0.45 | 0.0831 | -0.3904 | 0.0265 |
| | Particles 1000 vs Base (500) | -0.56 | 0.9996 | 0.33 | 0.3410 | -0.2380 | 0.0871 |
| | Particles 2000 vs Base (500) | -0.24 | 0.9995 | 0.31 | 0.6738 | -0.1916 | 0.1269 |
| | Res 0.05° vs Base (0.08°) | -0.72 | 0.9996 | 0.31 | 0.1917 | -0.2506 | 0.0542 |
| | Res 0.10° vs Base (0.08°) | -1.47 | 0.9997 | 0.39 | 0.0269 | -0.3729 | -0.0257 |
| PY_CO₂tot | Back 120 h vs Base (72 h) | -1.31 | 0.9992 | 0.44 | 0.0862 | -0.3819 | 0.0280 |
| | Back 96 h vs Base (72 h) | -1.33 | 0.9992 | 0.45 | 0.0916 | -0.3900 | 0.0322 |

| | | | | | | |
|---|---|---|---|---|---|---|
| Particles 1000 vs Base (500) | -0.52 | 0.9996 | 0.28 | 0.2983 | -0.2098 | 0.0683 |
| Particles 2000 vs Base (500) | -0.31 | 0.9995 | 0.29 | 0.5502 | -0.1864 | 0.1029 |
| Res 0.05° vs Base (0.08°) | -0.78 | 0.9996 | 0.31 | 0.1527 | -0.2541 | 0.0431 |
| Res 0.10° vs Base (0.08°) | -1.40 | 0.9997 | 0.35 | 0.0164 | -0.3394 | -0.0393 |

**Table S7.** Wintertime (12:00–16:00 LT) paired-day sensitivity of PY $CO_2$bio inferred fluxes to STILT parameter choices (n = 18). Variants (particle number, grid spacing, backward duration) are compared with the baseline (500 particles, 72 h, 0.08°). We report the absolute paired-day test–baseline difference, defined as $\Delta CO_2$bio = $CO_2$bio(variant) − $CO_2$bio(baseline), summarized by the paired-day mean ($\Delta$; $\mu$mol m$^{-2}$ s$^{-1}$) and its 95% confidence interval (CI95_lo, CI95_hi; $\mu$mol m$^{-2}$ s$^{-1}$). Because wintertime $CO_2$bio at PY is close to zero, percent differences are not shown. Across-run daily spread of $CO_2$bio—defined as the day-by-day standard deviation across the baseline and all variants—has median 0.045 and IQR 0.016–0.067 $\mu$mol m$^{-2}$ s$^{-1}$.

| metric | comparison | mean ($\Delta CO_2$bio) | ci95_lo | ci95_hi |
|---|---|---|---|---|
| PY_CO$_2$bio | Back 120 h vs Base (72 h) | −0.035 | −0.094 | 0.024 |
| | Back 96 h vs Base (72 h) | 0.003 | −0.057 | 0.063 |
| | Particles 1000 vs Base (500) | 0.005 | −0.031 | 0.041 |
| | Particles 2000 vs Base (500) | −0.009 | −0.052 | 0.033 |
| | Res 0.05° vs Base (0.08°) | −0.007 | −0.047 | 0.033 |
| | Res 0.10° vs Base (0.08°) | 0.010 | −0.043 | 0.063 |

Line 225: The statement that "footprint uncertainties are neglected under the assumption of unbiased atmospheric transport" seems to lack justification. Could the authors provide references supporting this choice or acknowledge its limitations?

Response:

We appreciate the reviewer's careful comment and agree that atmospheric transport (footprints) is an important uncertainty source and cannot be assumed error-free. Our original wording was not intended to claim "unbiased transport" but rather to indicate that we do not explicitly propagate footprint/transport uncertainty within the analytical error-propagation formulas used for Eqs. (12)–(14). We have revised the text to avoid any implication of unbiased transport, to clarify that transport uncertainty is not explicitly quantified in the analytical uncertainty terms, and to explicitly acknowledge this as a limitation.

Within a feasible scope, we additionally assess transport-model setup sensitivity by introducing a winter paired-day STILT sensitivity analysis at PY. In this test, we perturb key STILT configuration choices (particle number, grid resolution, and backward duration), recompute footprints, rerun the full flux-estimation workflow, and evaluate baseline–variant differences using effect sizes (percent differences), correlation (Pearson r), and paired statistical tests; the quantitative outcomes are reported in Sect. 3.5 and Fig. 11.

Finally, we now state explicitly that residual transport biases (e.g., winds and boundary-layer mixing) remain unquantified and may bias the inferred fluxes, thereby contributing to inventory–observation differences when benchmarking against bottom-up inventories, alongside emission-inventory uncertainty and the representativeness mismatch between footprint-weighted enhancements and unweighted inventory means. These revisions are implemented in Sect. 2.5.2 (Lines 306–312). This limitation is further discussed in Sect. 3.5 (Lines 668–670; Lines 678–680; Lines 686–696) and summarized in the Conclusions (Lines 754–756).

Fig 3A: Consider revising the x-axis label to "January–December" for clarity.

Response:

We appreciate the reviewer's suggestion. We have revised this (formerly Fig. 3) by replacing the numeric month ticks ("01–12") with month abbreviations ("Jan–Dec") to improve readability and clarity (Line 398).

[Figure]

**Figure 4.** (A) Variations in monthly mean $CO_2$ concentrations and (B) their correlations with Normalized Difference Vegetation Index (NDVI) for the (a–b) NS, (c–d) PY, and (e–f) CH stations. Error bars indicate ± 1 standard deviation (SD).

Lines 255–265: The urban–suburban differences highlighted here are insightful. Adding a figure to illustrate this spatial comparison would be valuable. Discussion of seasonal variations in this gradient would further enrich the analysis.

Response:

We appreciate the reviewer's constructive comment. To better visualize the urban–suburban spatial contrast and its seasonality, we added a new figure (Fig. 3) and expanded the discussion in Sect. 3.1 (Lines 355–357; Lines 368–385). Figure 3 summarizes the annual and seasonal mean gradients PY–NS and PY–CH (error bars: ±1 SE). The PY–NS gradient remains positive year-round but is smallest in winter (2.23 ppm), consistent with prevailing northerlies elevating $CO_2$ at the coastal NS site (Fig. 2). In contrast, PY–CH is more strongly seasonally modulated, peaking in winter (8.83 ppm) but reversing sign in summer (−2.86 ppm), consistent with more frequent southerly (marine-influenced) ventilation at PY (SW/SE sectors in Fig. 2) and seasonally enhanced $CO_2$ at CH linked to transport, biogenic and boundary-layer processes. Overall, these results highlight a seasonally displaced apparent $CO_2$ "dome" in this coastal setting, where the network maximum can occur outside the urban core; this coastal-specific feature becomes evident only when the spatial gradient is examined seasonally.

In addition, we integrated the gradient results into the broader mechanistic interpretation through targeted cross-references. In Sect. 3.1.1 (Lines 408–410), we clarify that the month-to-month south–north $CO_2$ pattern is consistent with the seasonal gradients in Fig. 3, including the winter weakening of the urban–coastal contrast and the summertime displacement of the urban–suburban gradient. We also added a cross-reference in Sect. 3.3 (Lines 536–538) linking the suburban summertime $CO_2$ maximum to the combined influence of coastal transport, biogenic exchange, and boundary-layer mixing, consistent with the seasonal gradient shift summarized in Fig. 3. For consistency, we aligned the Abstract and Conclusions with this new result (Lines 20–23; Lines 727–731).

[Figure]

**Figure 3.** Urban–suburban/coastal $CO_2$ gradients in Guangzhou. Annual and seasonal mean concentration differences ($\Delta CO_2$, ppm) between the urban site (PY) and the coastal site (NS) (PY–NS; blue bars) and between PY and the suburban forest site (CH) (PY–CH; orange bars). Seasons are defined as Spring (Mar–May), Summer (Jun–Aug), Autumn (Sep–Nov), and Winter (Dec–Feb). Error bars denote ± 1 standard error (SE).

Line 337: "with coastal > urban > suburban impacts" is not clear, please rephrase.

Response:

We thank the reviewer for pointing out this unclear phrasing. Here "coastal > urban > suburban impacts" refers to the magnitude of the SLBD-related $CO_2$ reduction (i.e., the ventilation/dispersion effect) across the three sites. We have rephrased the sentence to explicitly state that the SLBD-related $CO_2$ decrease is largest at the coastal site, intermediate at the urban site, and smallest at the suburban site (NS > PY > CH). We have revised this in Section 3.2 (Lines 481–484).

Line 425-429. This sentence is a little bit long and difficult to understand. Please rephrase and clarify.

Response:

We agree and have rewritten the sentence into several shorter sentences to improve clarity and readability. We have revised this in Section 3.5 (Lines 601–608).

Section 3.5: The reliability of the source analysis needs further strengthening. The current results are highly dependent on the accuracy of the prior emission inventory. However, significant discrepancies exist in fossil fuel emission estimates across different inventories, and emissions vary considerably

between years. Furthermore, the biogenic contribution is derived by subtracting the fossil fuel estimates. Therefore, it is strongly recommended that the authors assess the impact of using different inventories on the conclusions.

Response:

Thanks for this important point. We agree that bottom-up fossil-fuel $CO_2$ inventories can differ substantially across products and years. We therefore clarify that our $CO_2ff/CO_2bio$ partitioning is inventory-independent: $CO_2tot$ is inferred from observed concentration enhancements together with footprint-informed transport diagnostics, $CO_2ff$ is derived from observed CO enhancements using the PY-specific $R_{CO}$ relationship (determined from observed $\Delta CO/\Delta CO_2$ regression slopes), and $CO_2bio$ is diagnosed as the residual ($CO_2bio = CO_2tot - CO_2ff$). Emission inventories are not used as priors or constraints in the partitioning, but only for independent context and benchmarking. Consequently, inventory choice does not affect the inferred $CO_2tot/CO_2ff/CO_2bio$ or our main conclusions; it only affects the bottom-up benchmark range used for contextual comparison.

To address the reviewer's recommendation, we added a like-for-like cross-inventory benchmarking using same-year (2023) inventories at $0.1° \times 0.1°$, temporally disaggregated with the same hourly profiles, and compared winter-afternoon (12:00–16:00 LT) mean emissions aggregated over the winter footprint-defined sensitivity region (Fig. 12). This yields a wide inter-inventory spread (e.g., EDGAR vs. MEIC), which we now interpret explicitly as a plausibility envelope rather than a one-to-one validation target; the spread likely reflects differences in spatial disaggregation and point-source treatment that can be amplified at sub-provincial scales (Sect. 3.5; Lines 671–680).

We further clarify why inventory-based means and our observation-based estimates are not expected to match one-to-one: our $CO_2ff$ reflects a footprint-weighted receptor enhancement at PY (after background removal), whereas inventories provide unweighted gridded emission fields from which a domain-mean flux is computed over the sensitivity region. Because STILT footprints impose heterogeneous, transport-dependent weights and only grid cells with substantial influence contribute materially to the receptor signal (Fig. 12), the footprint-weighted effective mean can differ from the unweighted inventory mean, depending on the overlap between footprint patterns and spatially heterogeneous emissions (including hotspots) (Sect. 3.5; Lines 686–699). Regarding the residual formulation, we clarify that $CO_2bio$ is computed as ($CO_2tot - CO_2ff$), but $CO_2ff$ is not taken from inventories; it is inferred from observed CO using the site-specific $R_{CO}$ relationship. We explicitly

acknowledge that $CO_2bio$ inherits uncertainty from both $CO_2tot$ and $CO_2ff$ (Sect. 2.5.2; Lines 315–318; Sect. 3.5; Lines 616–623) and report bootstrap 95 % CIs for the summertime biogenic offset (Sect. 3.5; Lines 705–708). We further place the summertime offset in the context of independent coastal-urban estimates from the literature as an external consistency check (Sect. 3.5; Lines 709–715). As an additional robustness check, paired-day STILT configuration sensitivity tests show only small changes ($\leq$1.47 %) in inferred winter-afternoon $CO_2tot/CO_2ff$ across reasonable transport choices (Fig. 11; Tables S6–S7), and measurement/background-selection uncertainties contribute only a small fraction of the winter-afternoon $CO_2ff$ error budget (~3 %) (Sect. 3.5; Lines 641–668). These reliability assessments are summarized in the Conclusions (Lines 743–751).